# SMOOTHING FOR EXPONENTIAL FAMILY DYNAMICAL SYSTEMS

## ABSTRACT

State-space modeling is a powerful technique for the analysis of spatiotemporal structures of time series. However, when assumptions about linearity or Gaussianity are violated, statistical inference about the latent process is challenging. While variational inference can be used to approximate the posterior in these nonlinear or non-Gaussian settings, it is desirable to preserve the temporal structure of the true posterior in the variational approximation, while ensuring inference scales linearly in sequence length. We propose a new structured variational approximation that satisfies these desiderata. Furthermore, by generalizing to *exponential family dynamical systems*, we are able to develop decoupled second order inference algorithms that have simple updates, without increased computational complexity. Then, we extend our insights and develop the *auto-encoding backward factorized smoother*, making it easy to leverage modern deep learning tools. We compare against various inference algorithms for state-space models, and validate the theory presented through numerical experiments.

## 1 INTRODUCTION

Principled estimation of the unknown internal state evolution from noisy time series, known as *smoothing*, is fundamental to scientific discovery and engineering (Jazwinski, 2007; Pei et al., 2021; Koyama et al., 2010; Anderson & Moore, 1979; Douc et al., 2014; Durbin & Koopman, 2012). State-space modeling is a class of latent variable models that describe the generative process of internal states and observations, providing a spatiotemporal prior distribution as a scaffold for inference. Decades of research dedicated to solve the smoothing problem for non-Gaussian, nonlinear state-space models resulted in various approximate solutions including sampling based and variational approaches (Kitagawa, 1996; Douc et al., 2014; Turner & Sahani, 2011; Archer et al., 2015; Zhao et al., 2022). However, ensuring the meaningful temporal structure of the intractable but optimal posterior and ensuring inference scale linearly in sequence length is challenging.

We develop a myriad of novel variational smoothing algorithms that consider exponential family dynamical systems with arbitrary nonlinearity and arbitrary observation likelihood. While many works consider smoothing and filtering algorithms at the abstraction level of arbitrary probability distributions, usually, when the focus is narrowed, it is directly to models with additive Gaussian noise and Gaussian approximations. Using the duality of natural and mean parameter forms of exponential family distributions and analytical forms of optimal Kullback-Leibler divergences, we can leverage advantages of natural gradient descent for variational inference, and develop simplified algorithms.

We highlight our main contributions: **i)** We introduce a *prior-parameterized* backward factorization to the smoothing posterior, leading to a new evidence lower bound (ELBO). We make the new ELBO tractable by replacing the intractable distributions with their closest exponential-family approximations. **ii)** Using a Lagrange multiplier argument, we develop a smoothing algorithm capable of processing sequential data in parallel rather than in order; moreover, the induced stationary conditions for optimality reveals the dynamics

of natural parameters. **iii)** Drawing inspiration from the stationary conditions, we develop the *auto-encoding backward factorized smoother* (aBFS), allowing us to combine our insights with the modern deep learning toolbox.

## 2 BACKGROUND

### 2.1 BAYESIAN FILTERING AND SMOOTHING

Recursive Bayesian filtering and smoothing (Särkkä, 2013) provides a principled and algorithmically convenient way of computing the posterior distribution in state-space models,

$$p(\mathbf{y}_{1:T}, \mathbf{z}_{1:T}) = p(\mathbf{y}_1 \mid \mathbf{z}_1)p_{\boldsymbol{\theta}}(\mathbf{z}_1) \prod_{t=2}^{T} p(\mathbf{y}_t \mid \mathbf{z}_t)p_{\boldsymbol{\theta}}(\mathbf{z}_t \mid \mathbf{z}_{t-1}) \tag{1}$$

where $\mathbf{z}_t \in \mathbb{R}^L$ is the latent state that evolves according to the Markovian dynamics $p_{\boldsymbol{\theta}}(\mathbf{z}_t \mid \mathbf{z}_{t-1})$ parameterized by $\boldsymbol{\theta}$, and $\mathbf{y}_t \in \mathbb{R}^M$ are observations.

**Filtering** The goal of filtering is to recursively compute $\breve{p}(\mathbf{z}_t \mid \mathbf{y}_{1:t})$ – the posterior distribution over latent state at time $t$ given all data up to the present[1]. Given the current filtering posterior $\breve{p}(\mathbf{z}_t \mid \mathbf{y}_{1:t})$, we can forecast the future state via the *predictive distribution*

$$\bar{p}(\mathbf{z}_{t+1} \mid \mathbf{y}_{1:t}) = \mathbb{E}_{\breve{p}(\mathbf{z}_t \mid \mathbf{y}_{1:t})} \left[ p_{\boldsymbol{\theta}}(\mathbf{z}_{t+1} \mid \mathbf{z}_t) \right] \tag{p.1}$$

where $^-$ is a decorator used to denote a predictive distribution or parameter. After observing $\mathbf{y}_{t+1}$, we update our prediction through Bayes' rule,

$$\breve{p}(\mathbf{z}_{t+1} \mid \mathbf{y}_{1:t+1}) \propto p(\mathbf{y}_{t+1} \mid \mathbf{z}_{t+1})\bar{p}(\mathbf{z}_{t+1} \mid \mathbf{y}_{1:t}) \tag{p.2}$$

These recursive updates beginning with, $p(\mathbf{z}_1)$, produce a series of predictive and filtering distributions.

**Smoothing** Smoothing is to update the belief of latent states given all the observations. After the filtering pass has produced necessary intermediary quantities (Kitagawa, 1996), the smoothing distribution $p(\mathbf{z}_t \mid \mathbf{y}_{1:T})$ can be recursively computed backwards in time, starting with $\breve{p}(\mathbf{z}_T \mid \mathbf{y}_{1:T})$. The procedure also comprises two steps,

$$p(\mathbf{z}_t \mid \mathbf{z}_{t+1}, \mathbf{y}_{1:T}) = \frac{p_{\boldsymbol{\theta}}(\mathbf{z}_{t+1} \mid \mathbf{z}_t)\breve{p}(\mathbf{z}_t \mid \mathbf{y}_{1:t})}{\mathbb{E}_{\breve{p}(\mathbf{z}_t \mid \mathbf{y}_{1:t})} \left[ p_{\boldsymbol{\theta}}(\mathbf{z}_{t+1} \mid \mathbf{z}_t) \right]} \tag{p.3}$$

$$p(\mathbf{z}_t \mid \mathbf{y}_{1:T}) = \mathbb{E}_{p(\mathbf{z}_{t+1} \mid \mathbf{y}_{1:T})} \left[ p(\mathbf{z}_t \mid \mathbf{z}_{t+1}, \mathbf{y}_{1:T}) \right] \tag{p.4}$$

The Kalman filter and smoother recover the exact posterior (Jazwinski, 2007; Särkkä, 2013) for a linear Gaussian state-space model (LGSSM) (App. B). However, for more general cases, the filtering and smoothing distributions are often intractable.

### 2.2 VARIATIONAL INFERENCE FOR STATE-SPACE MODELS

Variational inference is one way to deal with the intractable posterior by finding an approximation $q(\mathbf{z}_{1:T}) \approx p(\mathbf{z}_{1:T} \mid \mathbf{y}_{1:T})$ (Blei et al., 2017). Usually, $q$ is chosen to be a member of a parametric family of distributions, $\mathcal{Q}$, whose parameters are found by maximizing the evidence lower bound (ELBO),

$$\mathcal{L}(q) = \sum_{t=1}^{T} \mathbb{E}_{q(\mathbf{z}_t)} \left[ \log p(\mathbf{y}_t \mid \mathbf{z}_t) \right] - \mathbb{D}_{\mathrm{KL}}(q(\mathbf{z}_{1:T}) \| p_{\boldsymbol{\theta}}(\mathbf{z}_{1:T})) \leq \log p(\mathbf{y}_{1:T}). \tag{2}$$

**Exponential family dynamics** Many existing works on variational inference for state-space models narrow their focus to dynamical systems driven by additive Gaussian noise, and that naturally guides the focus to Gaussian variational approximations (Karl et al., 2016; Krishnan et al., 2016; Fraccaro et al., 2017; Archer

---

[1] $\breve{}$ is a decorator to remind us this distribution is conditioned on past and present data, see App. A for nomenclature

et al., 2014; Campbell et al., 2021; Klushyn et al., 2021). However, by keeping our level of abstraction to exponential family distributions, it becomes easier to understand the structure of an optimal variational posterior. This also allows for a variety of stochasticity to which many well-known dynamics are special cases, simplifies the analysis, and makes it possible to exploit the information geometry of exponential families by using natural gradient descent (Amari, 1998). We consider exponential family dynamical systems (Dowling et al., 2023) defined by,

───── exponential family dynamics ─────

$$p(\mathbf{z}_{t+1} \mid \mathbf{z}_t) = h(\mathbf{z}_{t+1}) \exp\left(t(\mathbf{z}_{t+1})^\top \boldsymbol{\lambda_\theta}(\mathbf{z}_t) - A\left(\boldsymbol{\lambda_\theta}(\mathbf{z}_t)\right)\right) \tag{3}$$

where $t(\mathbf{z}_{t+1})$ are the sufficient statistics, $A(\cdot)$ is the log-partition function, $h(\cdot)$ the base measure, and $\boldsymbol{\lambda_\theta}(\mathbf{z}_t)$ captures the dynamics by describing how $\mathbf{z}_t$ is transformed into the natural parameters of $\mathbf{z}_{t+1}$. In App. B, show how a LGSSM can be written in this form. Without loss of generality, we consider only minimal exponential family distributions (Wainwright & Jordan, 2008) for which there exists a one-to-one mapping between the natural and mean parameters s.t. $\boldsymbol{\mu} := \mathbb{E}_{p(\mathbf{z}\mid\boldsymbol{\lambda})}[t(\mathbf{z})] = \nabla A(\boldsymbol{\lambda})$. This choice (1) eliminates the ambiguity in notation, and lets us write $\boldsymbol{\mu_\theta}(\mathbf{z}_t) = \nabla_{\boldsymbol{\lambda_\theta}} A(\boldsymbol{\lambda_\theta}(\mathbf{z}_t))$, (2) allows the Fisher information to be written as the Hessian of the log-partition function s.t. $\mathcal{I}(\boldsymbol{\lambda}) = \nabla^2 A(\boldsymbol{\lambda})$, and (3) lets us use the duality connecting mean and natural parameters through the relation $\mathcal{I}(\boldsymbol{\lambda})^{-1}\nabla_{\boldsymbol{\lambda}} g(\boldsymbol{\lambda}) = \nabla_{\boldsymbol{\mu}} g(\boldsymbol{\lambda})$ (Khan & Nielsen, 2018).

**Forward and reverse KL divergences**     If $q(\mathbf{z})$ and $p(\mathbf{z})$ are in the same exponential family, denoted $\mathcal{Q}$, and have natural parameters $\boldsymbol{\lambda}$ and $\boldsymbol{\lambda}_0$ respectively, then at a stationary point (optima) of $\mathbb{D}_{\mathrm{KL}}(q(\mathbf{z})\|p(\mathbf{y}\mid\mathbf{z})p(\mathbf{z})/p(\mathbf{y}))$ with respect to $\boldsymbol{\lambda}$, the parameters of variational approximation satisfy the implicit equation (Khan & Nielsen, 2018)

$$\boldsymbol{\lambda}^\star = \boldsymbol{\lambda}_0 + \nabla_{\boldsymbol{\mu}^\star}\mathbb{E}_{q(\mathbf{z};\boldsymbol{\lambda}^\star)}\left[\log p(\mathbf{y}\mid\mathbf{z})\right] := \boldsymbol{\lambda}_0 + \tilde{\boldsymbol{\lambda}}(\mathbf{y}, \boldsymbol{\lambda}^\star) \tag{4}$$

where we have defined $\tilde{\boldsymbol{\lambda}}(\mathbf{y}, \boldsymbol{\lambda}) := \nabla_{\boldsymbol{\mu}}\mathbb{E}_{q(\mathbf{z};\boldsymbol{\lambda})}\left[\log p(\mathbf{y}\mid\mathbf{z})\right]$. This lends itself to the interpretation that at a stationary point of the ELBO, the natural parameters of the variational approximation resemble a conjugate Bayesian update (Khan & Lin, 2017); if the solution is not implicit so that we can write $\tilde{\boldsymbol{\lambda}}(\mathbf{y}, \boldsymbol{\lambda}) = \tilde{\boldsymbol{\lambda}}(\mathbf{y}) -$ which happens if the prior is conjugate to the likelihood – then, one natural gradient step on the ELBO is equivalent to the exact Bayesian conjugate update.

It is also well known that a stationary point of $\mathbb{D}_{\mathrm{KL}}(p(\mathbf{z})\|q(\mathbf{z}))$ with respect to $\boldsymbol{\lambda}$ the moment matching condition is satisfied so that, $\boldsymbol{\mu} = \mathbb{E}_{p(\mathbf{z})}[t(\mathbf{z})]$ where $t(\mathbf{z})$ are the sufficient statistics of $q \in \mathcal{Q}$ (Minka, 2001). However, a lesser known fact is that at a stationary point of

$$\mathbb{D}_{\mathrm{KL}}\left(\mathbb{E}_{p(\mathbf{z}_1)}\left[p(\mathbf{z}_2 \mid \mathbf{z}_1)\right]\big\|\, q(\mathbf{z}_2)\right) \tag{5}$$

when $p(\mathbf{z}_2 \mid \mathbf{z}_1) \in \mathcal{Q}$ and has natural parameters, $\boldsymbol{\lambda}_{2|1}(\mathbf{z}_1)$, the mean parameters, $\boldsymbol{\mu}_2^\star$, satisfy $\boldsymbol{\mu}_2^\star = \mathbb{E}_{p(\mathbf{z}_1)}\left[\boldsymbol{\mu}_{2|1}(\mathbf{z}_1)\right]$ which we show in App. D. Said in words – the best variational approximation to the marginalized (mixture) distribution is the one whose mean parameters are the expected mean parameters of the conditional mean mapping under the prior. These observations suggest that in order to approximate a posterior, the forward KL may be a favorable objective. On the other hand, if we are trying to approximate an intractable marginalization, then the reverse KL is more convenient.

## 3   A FRESH LOOK AT THE STATE-SPACE MODEL ELBO

An interesting fact is that, while the states *a priori* form a first order Markov chain, they *a posteriori* form a first order Markov chain time-reversely. With the freedom of choosing the structure of the variational approximation, we are allowed to exploit this duality by factoring the variational approximation in accordance

with the true posterior as a time-reversed Markov chain (Pfrommer & Matni, 2022; Campbell et al., 2021),

$$q(\mathbf{z}_{1:T}) = q(\mathbf{z}_T) \prod_{t=1}^{T-1} q(\mathbf{z}_t \mid \mathbf{z}_{t+1}) \tag{6}$$

Plugging the backward factorization into Eq. (2) leads to a simplified ELBO,

$$\mathcal{L}_B(q) = \sum_{t=1}^{T} \mathbb{E}_{q(\mathbf{z}_t)} \left[\log p(\mathbf{y}_t \mid \mathbf{z}_t)\right] - \mathbb{E}_{q(\mathbf{z}_{1:T})} \left[\log \frac{q(\mathbf{z}_T)}{p_{\boldsymbol{\theta}}(\mathbf{z}_1)} \prod_{t=1}^{T-1} \frac{q(\mathbf{z}_t \mid \mathbf{z}_{t+1})}{p_{\boldsymbol{\theta}}(\mathbf{z}_{t+1} \mid \mathbf{z}_t)}\right] \tag{7}$$

This bound would be sufficient to directly amortize inference, e.g., like Campbell et al. (2021) and learn an amortized backward transition distribution, $q_{\boldsymbol{\phi}}(\mathbf{z}_t \mid \mathbf{z}_{t+1})$, parameterized by a neural network with weights $\boldsymbol{\phi}$. However, this would require learning an entire separate amortization network which relies on a good learning of the SSM (with unknown parameters), and usually come with two drawbacks: underutilized *a priori* dynamics (a blackbox amortization network does not use explicitly the structure of dynamics.) and inefficient training of dynamics model.

**Prior-parameterized backward factorization**   Our key insight, is to elect to factor the variational approximation by incorporating the model of the dynamics, $p_{\boldsymbol{\theta}}(\mathbf{z}_{t+1} \mid \mathbf{z}_t)$, so that it factors similar to the true posterior backward distribution, $p(\mathbf{z}_t \mid \mathbf{z}_{t+1}, \mathbf{y}_{1:T})$, in Eq. (p.3). With this design choice,

> prior-parameterized variational approximation
> $$q(\mathbf{z}_t \mid \mathbf{z}_{t+1}) = \frac{p_{\boldsymbol{\theta}}(\mathbf{z}_{t+1} \mid \mathbf{z}_t) q(\mathbf{z}_t)}{\mathbb{E}_{q(\mathbf{z}_t)}\left[p_{\boldsymbol{\theta}}(\mathbf{z}_{t+1} \mid \mathbf{z}_t)\right]} \tag{8}$$

This factorization leads to a variational posterior with $\mathcal{O}(T)$ (we omit the other dimensions for simplicity) parameters to specify the marginals of the variational approximation, $q(\mathbf{z}_t)$ for $t \in [1:T]$; it scales like a mean-field approximation but without the assumption of independence (Turner & Sahani, 2011; Opper & Archambeau, 2009). Plugging this factorization into Eq. (7) results in convenient cancellations so that,

$$\mathcal{L}_B(q) = \sum_{t=1}^{T} \mathbb{E}_{q_t} \left[\log p(\mathbf{y}_t \mid \mathbf{z}_t)\right] - \sum_{t=2}^{T} \mathbb{D}_{\mathrm{KL}}\big(q_t \big|\big| \mathbb{E}_{q_{t-1}}\left[p_{t|t-1}\right]\big) - \mathbb{D}_{\mathrm{KL}}(q_1 || p_1) \tag{9}$$

**Remark**   For a forward factorization, $q(\mathbf{z}_{1:T}) = q(\mathbf{z}_1) \prod q(\mathbf{z}_{t+1} \mid \mathbf{z}_t)$, the corresponding ELBO is,

$$\mathcal{L}_F(q) = \sum_{t=1}^{T} \mathbb{E}_{q_t} \left[\log p(\mathbf{y}_t \mid \mathbf{z}_t)\right] - \sum_{t=2}^{T} \mathbb{E}_{q_{t-1}} \left[\mathbb{D}_{\mathrm{KL}}\big(q_{t|t-1} \big|\big| p_{t|t-1}\big)\right] - \mathbb{D}_{\mathrm{KL}}(q_1 || p_1) \tag{10}$$

This reveals an interesting dichotomy between $\mathcal{L}_F(q)$ and $\mathcal{L}_B(q)$ – the forward factorization leads to $\mathcal{L}_F(q)$ having terms that are expected KLs of conditional distributions, but the backward factorization leads to $\mathcal{L}_B(q)$ having terms that are KLs of expected conditional distributions. The advantage of $\mathcal{L}_F(q)$ is that, if the computation of $q(\mathbf{z}_t \mid \mathbf{z}_{t-1})$ is amortized, sampling trajectories from the posterior is straightforward (Krishnan et al., 2016; Karl et al., 2016).

The computation of KL terms of $\mathcal{L}_B(q)$ tend to be not as easy since $\mathbb{E}_{q_{t-1}}\left[p_{t|t-1}\right]$ does not necessarily belong to the same exponential family and thus the KL term will require a stochastic estimate. Inspired by this observation, we will proceed by considering an approximation to this term for the sake of analytic tractability. The proceeding approximation will have two favorable properties: i) we choose the best approximation in the same exponential family under the forward KL (See 2.2), and ii) in linear and Gaussian cases, it attains the exact term.

## 3.1 Tractable $\mathcal{L}_B(q)$

Now, to facilitate further mathematical tractability, we restrict $q(\mathbf{z}_t)$ to be in the same exponential family that $p_{\boldsymbol{\theta}}(\mathbf{z}_{t+1} \mid \mathbf{z}_t)$ belongs to, i.e.,

$$q(\mathbf{z}_t) = h(\mathbf{z}_t) \exp\left(t(\mathbf{z}_t)^\top \boldsymbol{\lambda}_t - A(\boldsymbol{\lambda}_t)\right) \tag{11}$$

As a result of the factorization chosen in Eq. (8), this choice does **not** confine the joint variational posterior $q(\mathbf{z}_{1:T})$ to be a member of the same exponential family.

**Plug-in predictive distribution** The predictive distribution, $\mathbb{E}_{q(\mathbf{z}_{t-1})}\left[p_{\boldsymbol{\theta}}(\mathbf{z}_t \mid \mathbf{z}_{t-1})\right]$, is generally intractable, and so evaluating the KL terms in Eq. (8) cannot be done in closed form. Though it is possible to employ Monte-Carlo estimates, the resulting stochastic gradient will be costly or have prohibitively large variance.

Fortunately, we can circumvent this problem by replacing the predictive distribution with an approximation that allows the KL term and its gradients to be calculated in closed form. Following this mindset, we find the closest projection of $\mathbb{E}_{q(\mathbf{z}_{t-1})}\left[p_{\boldsymbol{\theta}}(\mathbf{z}_t \mid \mathbf{z}_{t-1})\right]$ onto the family $\mathcal{Q}$ through the forward KL,

───────────────────── plug-in predictive distribution ─────────────────────

$$\bar{q}^\star(\mathbf{z}_t) = \underset{\bar{q} \in \mathcal{Q}}{\arg\min} \; \mathbb{D}_{\mathrm{KL}}\left(\mathbb{E}_{q(\mathbf{z}_{t-1})}\left[p_{\boldsymbol{\theta}}(\mathbf{z}_t \mid \mathbf{z}_{t-1})\right] \,\middle\|\, \bar{q}(\mathbf{z}_t)\right), \qquad \bar{\boldsymbol{\mu}}_t^\star = \mathbb{E}_{q(\mathbf{z}_{t-1})}\left[\boldsymbol{\mu}_{\boldsymbol{\theta}}(\mathbf{z}_{t-1})\right] \tag{12}$$

so that, $\bar{q}(\mathbf{z}_t) \approx \mathbb{E}_{q(\mathbf{z}_{t-1})}\left[p_{\boldsymbol{\theta}}(\mathbf{z}_t \mid \mathbf{z}_{t-1})\right]$ with mean parameters $\bar{\boldsymbol{\mu}}_t = \mathbb{E}_{q(\mathbf{z}_{t-1})}\left[\boldsymbol{\mu}_{\boldsymbol{\theta}}(\mathbf{z}_{t-1})\right]$ is a 'plug-in' approximation to the predictive distribution. One nice feature of this approximation can be seen when we consider the most common state-space model, the LGGSM. To be exact, consider when the dynamics are given by

$$p_{\boldsymbol{\theta}}(\mathbf{z}_t \mid \mathbf{z}_{t-1}) = \mathcal{N}(\mathbf{z}_t \mid \mathbf{A}\mathbf{z}_{t-1}, \mathbf{Q}) \tag{13}$$

In this case, the mean parameter mapping of the dynamics is affine in the sufficient statistics, so we can write $\boldsymbol{\mu}_{\boldsymbol{\theta}}(\mathbf{z}_{t-1}) = \mathbf{G}t(\mathbf{z}_{t-1}) + \mathbf{g}$ where $t(\mathbf{z}_{t-1}) = \left(\mathbf{z}_{t-1}^\top \quad -\frac{1}{2}\mathrm{vec}(\mathbf{z}_{t-1}\mathbf{z}_{t-1}^\top)^\top\right)^\top$ and,

$$\mathbf{G} = \begin{pmatrix} \mathbf{A} & \mathbf{0} \\ \mathbf{0} & \mathbf{A} \otimes \mathbf{A} \end{pmatrix} \quad \mathbf{g} = -\frac{1}{2}\begin{pmatrix} \mathbf{0} \\ \mathbf{Q} \end{pmatrix}. \tag{14}$$

Then, if we had that $q(\mathbf{z}_{t-1}) = \mathcal{N}(\mathbf{z}_{t-1} \mid \mathbf{m}_{t-1}, \mathbf{P}_{t-1})$, we can perform the exact marginalization

$$\mathbb{E}_{q(\mathbf{z}_{t-1})}\left[p_{\boldsymbol{\theta}}(\mathbf{z}_t \mid \mathbf{z}_{t-1})\right] = \mathcal{N}(\mathbf{z}_t \mid \mathbf{A}\mathbf{m}_{t-1}, \mathbf{A}\mathbf{P}_{t-1}\mathbf{A}^\top + \mathbf{Q}) \coloneqq \bar{q}(\mathbf{z}_t) \tag{15}$$

in which case the mean parameters of $\bar{q}(\mathbf{z}_t)$ would be

$$\bar{\boldsymbol{\mu}}_t = \begin{pmatrix} \mathbf{A}\mathbf{m}_{t-1} \\ -\frac{1}{2}(\mathbf{A} \otimes \mathbf{A})\mathrm{vec}(\mathbf{P}_{t-1} + \mathbf{m}_{t-1}\mathbf{m}_{t-1}^\top) - \frac{1}{2}\mathrm{vec}(\mathbf{Q}) \end{pmatrix} \tag{16}$$

which is the same result had we directly set $\bar{\boldsymbol{\mu}}_t = \mathbb{E}_{q(\mathbf{z}_{t-1})}\left[\mathbf{G}t(\mathbf{z}_{t-1}) + \mathbf{g}\right]$.

Now that $q_t$ and $\bar{q}_t$ belong in the same exponential family, the natural gradients of the KL are conveniently given by the difference of natural parameters (Khan & Lin, 2017), so that $\nabla_{\boldsymbol{\mu}_t}\mathbb{D}_{\mathrm{KL}}(q(\mathbf{z}_t)\|\bar{q}(\mathbf{z}_t)) = \boldsymbol{\lambda}_t - \bar{\boldsymbol{\lambda}}_t$. As a result, natural gradient ascent on $\mathcal{L}_B(q)$ is convenient since the Fisher information does not have to be computed. Replacing the intractable marginalization with $\bar{q}(\mathbf{z}_t)$ results in an approximate ELBO,

$$\widehat{\mathcal{L}}_B(\boldsymbol{\lambda}_{1:T}) = \sum_{t=1}^{T} \mathbb{E}_{q(\mathbf{z}_t)}\left[\log p(\mathbf{y}_t \mid \mathbf{z}_t)\right] - \mathbb{D}_{\mathrm{KL}}(q(\mathbf{z}_t)\|\bar{q}(\mathbf{z}_t)) \tag{17}$$

where $\bar{q}(\mathbf{z}_1) = p(\mathbf{z}_1)$, so that $\bar{\boldsymbol{\mu}}_1$ are the mean parameters of $p(\mathbf{z}_1)$, and $\bar{\boldsymbol{\mu}}_t = \mathbb{E}_{q_{t-1}}\left[\boldsymbol{\mu}_{\boldsymbol{\theta}}(\mathbf{z}_t)\right]$ are the mean parameters of $\bar{q}(\mathbf{z}_t)$ for $t \in [2:T]$; and although we call it an ELBO, it is important to keep in mind it is not necessarily a lower bound on the evidence because of the plug-in predictive approximation. At a stationary

point of $\widehat{\mathcal{L}}_B$, the natural parameters of the approximate marginal, $\boldsymbol{\lambda}_t^\star$, satisfy the implicit relationship,

$$\boldsymbol{\lambda}_t^\star = \bar{\boldsymbol{\lambda}}_t^\star + \nabla_{\boldsymbol{\mu}_t^\star} \mathbb{E}_{q(\mathbf{z}_t | \boldsymbol{\lambda}^\star)} \left[ \log p(\mathbf{y}_t \mid \mathbf{z}_t) \right] + \left[ \nabla_{\boldsymbol{\mu}_t^\star} \mathbb{E}_{q(\mathbf{z}_t | \boldsymbol{\lambda}_t^\star)} \left[ \boldsymbol{\mu}_{\boldsymbol{\theta}}(\mathbf{z}_t) \right] \right] \mathcal{I}(\bar{\boldsymbol{\lambda}}_{t+1}^\star)^{-1} (\boldsymbol{\mu}_{t+1}^\star - \bar{\boldsymbol{\mu}}_{t+1}^\star)$$

$$:= \mathfrak{F}(\boldsymbol{\lambda}_{t-1}^\star) + \tilde{\boldsymbol{\lambda}}(\mathbf{y}_t, \boldsymbol{\lambda}_t^\star) + \mathbf{M}_t(\boldsymbol{\lambda}_t^\star) \mathcal{I}(\mathfrak{F}(\boldsymbol{\lambda}_t^\star))^{-1} \left( \boldsymbol{\mu}_{t+1}^\star - \mathfrak{DL}^{-1}(\boldsymbol{\lambda}_t^\star) \right) \tag{18}$$

where $\mathfrak{F} : \boldsymbol{\lambda}_t \mapsto \bar{\boldsymbol{\lambda}}_{t+1}$, is a composition of transformations $\boldsymbol{\lambda}_t \xrightarrow{\mathfrak{L}^{-1}} \boldsymbol{\mu}_t \xrightarrow{\mathfrak{D}} \bar{\boldsymbol{\mu}}_{t+1} \xrightarrow{\mathfrak{L}} \bar{\boldsymbol{\lambda}}_{t+1}$, which push the mean parameters through the dynamics. For convenience, we write $\mathfrak{F} = \mathfrak{L} \circ \mathfrak{D} \circ \mathfrak{L}^{-1}$ where

$$\mathfrak{L}^{-1}(\boldsymbol{\lambda}) = \nabla A(\boldsymbol{\lambda}) = \boldsymbol{\mu} \qquad \mathfrak{D}(\boldsymbol{\mu}) = \mathbb{E}_{q(\mathbf{z} | \boldsymbol{\lambda}(\boldsymbol{\mu}))} \left[ \boldsymbol{\mu}_{\boldsymbol{\theta}}(\mathbf{z}) \right] \qquad \mathfrak{L}(\boldsymbol{\mu}) = \nabla A^*(\boldsymbol{\mu}) = \boldsymbol{\lambda}. \tag{19}$$

and we define the Jacobian $\mathbf{M}_t(\boldsymbol{\lambda}_t) := \nabla_{\boldsymbol{\mu}_t} \mathbb{E}_{q_t} \left[ \boldsymbol{\mu}_{\boldsymbol{\theta}}(\mathbf{z}_t) \right]$.

**Constrained plug-in predictive ELBO** The intricate dependence of $\boldsymbol{\lambda}_t$ on its neighbors, $\boldsymbol{\lambda}_{t-1}$ and $\boldsymbol{\lambda}_{t+1}$, complicate optimizing $\widehat{\mathcal{L}}_B$. To ameliorate the difficulty, we first convert it to a constrained problem

$$\text{maximize} \quad \widehat{\mathcal{L}}_{B,C}(\boldsymbol{\lambda}_{1:T}, \bar{\boldsymbol{\mu}}_{2:T}) := \left[ \sum_{t=1}^T \mathbb{E}_{q(\mathbf{z}_t)} \left[ \log p(\mathbf{y}_t \mid \mathbf{z}_t) \right] - \mathbb{D}_{\text{KL}}(q(\mathbf{z}_t) \| \bar{q}(\mathbf{z}_t)) \right] \tag{20}$$

$$\text{subject to} \quad \bar{\boldsymbol{\mu}}_t = \mathbb{E}_{q(\mathbf{z}_{t-1})} \left[ \boldsymbol{\mu}_{\boldsymbol{\theta}}(\mathbf{z}_{t-1}) \right], \quad t = 2, \dots, T \tag{21}$$

where we have made $\bar{\boldsymbol{\mu}}_{2:T}$ free variables that must equal the plug-in predictive mean parameters of Eq. (12). The stationary points (optima) of the constrained problem coincide with those of $\widehat{\mathcal{L}}_B$.

## 3.2 Optimizing $\widehat{\mathcal{L}}_{B,C}$

We propose two approaches to optimize the constrained plug-in ELBO. One uses Lagrangian multipliers, and the other uses the variational autoencoding (VAE) framework (Kingma & Welling, 2014) to amortize inference.

**Lagrangian of the ELBO** For the constrained optimization, we employ a Lagrange multiplier argument. Letting $\boldsymbol{\nu}_{2:T}$ be the Lagrange multipliers for the constraint, the Lagrangian is

$$\widehat{\mathcal{L}}_{B,U}(\boldsymbol{\lambda}_{1:T}, \bar{\boldsymbol{\mu}}_{2:T}, \boldsymbol{\nu}_{2:T}) = \sum_{t=1}^T \mathbb{E}_{q_t} \left[ \log p(\mathbf{y}_t \mid \mathbf{z}_t) \right] - \mathbb{D}_{\text{KL}}(q_t \| \bar{q}_t) - \boldsymbol{\nu}_t^\top \left( \bar{\boldsymbol{\mu}}_t - \mathbb{E}_{q_{t-1}} \left[ \boldsymbol{\mu}_{\boldsymbol{\theta}}(\mathbf{z}_{t-1}) \right] \right) \tag{22}$$

At a stationary point of this objective, the natural parameters and Lagrange multipliers are related as,

decoupled backward factorized smoother (dBFS) stationary conditions

$$\boldsymbol{\lambda}_t^\star = \mathfrak{F}(\boldsymbol{\lambda}_{t-1}^\star) + \tilde{\boldsymbol{\lambda}}_t(\mathbf{y}_t, \boldsymbol{\lambda}_t^\star) + \mathbf{M}_t(\boldsymbol{\lambda}_t^\star)^\top \boldsymbol{\nu}_{t+1}^\star \tag{23}$$

$$\boldsymbol{\nu}_t^\star = \mathcal{I}(\bar{\boldsymbol{\lambda}}_t^\star)^{-1} (\boldsymbol{\mu}_t^\star - \bar{\boldsymbol{\mu}}_t^\star) \tag{24}$$

From the stationary conditions, the optimal variational parameters evolve forward in time according to dynamics given by $\mathfrak{F}(\cdot)$, and the future information flows backwards through the Lagrange multipliers. Though an analytical solution is often intractable, a numerical solution satisfying the stationarity conditions is achievable via a dual ascent method Nocedal & Wright (1999) – we present an example implementation in App. H.1 Fig. 5a. For the inner loop, both $\boldsymbol{\lambda}_t$ and $\bar{\boldsymbol{\lambda}}_t$ are updated through natural gradient descent. By decoupling neighboring parameters, the Lagrange multipliers make it possible to optimize the variational parameters in parallel.

**Natural parameter amortization** The stationary condition, Eq. (23), shows how the variational posterior integrates information from the past, present, and future. In the spirit of VAEs, this suggests the interesting possibility of amortizing posterior computation by learning inference networks that approximate the posterior as,

auto-encoding backward factorized smoother (aBFS)

$$\boldsymbol{\lambda}_t = \mathfrak{F}(\boldsymbol{\lambda}_{t-1}) + \tilde{\boldsymbol{\lambda}}_{\boldsymbol{\phi}}(\mathbf{y}_t) + \mathbf{u}_{\boldsymbol{\phi}}(\mathbf{y}_{t+1:T}), \tag{25}$$

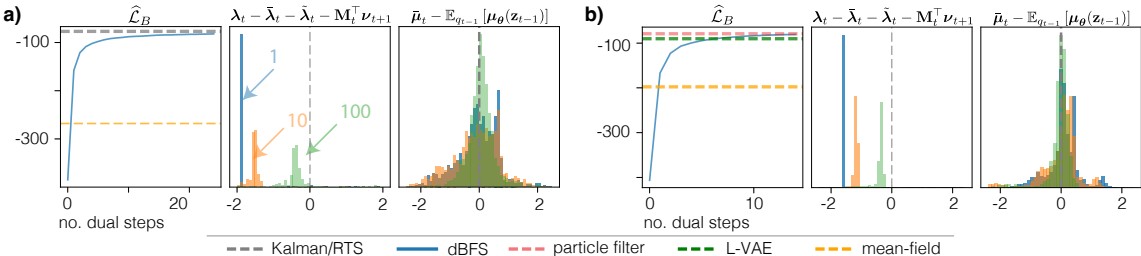

Figure 1: dBFS convergence on **a)** linear and **b)** nonlinear (Van der Pol) latent dynamics. In each of a) and b), we show (left) the ELBO as a function of the number of dual variable updates; (middle) histogram of the difference between current natural parameters and the optimal point (Eq. (23)) at dual step 1, 10 and 100; (right) histogram of the difference between inferred predictive mean parameters and the expected mean parameters under the posterior (Eq. (24))

where $\tilde{\boldsymbol{\lambda}}_{\boldsymbol{\phi}}(\cdot)$, and $\mathbf{u}_{\boldsymbol{\phi}}(\cdot)$ are neural networks with weights, $\boldsymbol{\phi}$, to be trained by stochastic back-propagation through Eq. (17) (Rezende et al., 2014). Importantly, Eq. (25) isn't the only way we could amortize inference – we could also consider variants like $\boldsymbol{\lambda}_t = \mathfrak{F}(\boldsymbol{\lambda}_{t-1}) + \tilde{\boldsymbol{\lambda}}_{\boldsymbol{\phi}}(\mathbf{y}_t, \bar{\boldsymbol{\lambda}}_t) + \mathbf{u}_{\boldsymbol{\phi}}(\mathbf{y}_{t+1:T})$ which may be better able to capture an implicit relationship like Eq. (23), and narrow the amortization gap that arises from the inability of the inference network to capture dependencies in the graphical model (Cremer et al., 2018).

The natural parameters required to evaluate $\widehat{\mathcal{L}}_B(\boldsymbol{\phi}, \boldsymbol{\theta})$ can be computed recursively through Eq. (25). Once computed, stochastic backpropagation can be used to learn the parameters of the inference networks, without sacrificing tight control over the structure of the variational approximation. For example, a simple, but expressive inference network could be constructed by parameterizing $\mathbf{u}_{\boldsymbol{\phi}}(\cdot)$ as an RNN running backward in time and $\tilde{\boldsymbol{\lambda}}_{\boldsymbol{\phi}}(\cdot)$ as a convolutional neural network (CNN) or multilayer perceptron (MLP). We summarize the algorithm in Fig. 5b.

For aBFS, another unique advantage is that the backward encoder can be made to depend on the observations, $\mathbf{y}_{t+1:T}$, through their natural parameter encodings (given by passing them through $\boldsymbol{\lambda}_{\boldsymbol{\phi}}(\cdot)$); for many problems, the dimensionality of the latent space is lower than the observation space, so it is advantageous that the backward encoder can depend on the low dimensional natural parameter encodings instead of the high-dimensional raw data.

**Learning the SSM parameters** As noted in various works, learning a model of the dynamics that facilitates accurate long term prediction is difficult since the parameters of the dynamics are only backpropagated through one time step when evaluating the ELBO (Karl et al., 2016; Hafner et al., 2019). For standard variational inference, this can be mitigated by considering k-step objectives that force samples to traverse the dynamics for multiple time-steps. For aBFS, time points can effectively be masked by setting $\tilde{\boldsymbol{\lambda}}(\cdot)$ to $\mathbf{0}$, so that the natural parameters for the posterior of a latent state, $\mathbf{z}_t$, associated with a masked observation would be calculated as $\boldsymbol{\lambda}_t = \mathfrak{F}(\boldsymbol{\lambda}_{t-1}) + \mathbf{u}_{\boldsymbol{\phi}}(\mathbf{y}_{t+1:T})$. While this is like the masking mechanism in Zhao & Linderman (2023), in that work it only applies to linear dynamical systems. In our case, by paying the price of a backward encoder, we are able to realize the same masking mechanism applied to learning nonlinear dynamics. Other parameters, such as those of the likelihood, can be learned either end-to-end or by using variational expectation maximization (EM) (Turner & Sahani, 2011).

## 4 RELATED WORKS

Contemporary works like Fraccaro et al. (2017); Krishnan et al. (2016); Klushyn et al. (2021); Chung et al. (2015); Kaiser et al. (2019); Karl et al. (2016) adopt VAEs for state estimation and system identification of

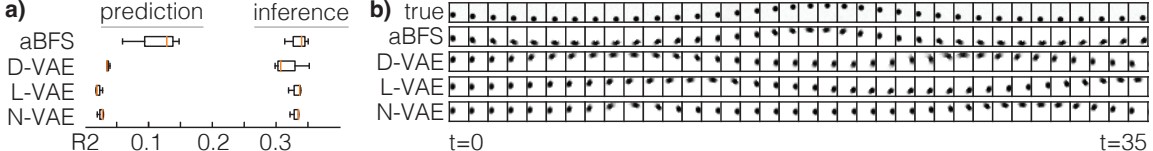

Figure 2: Inference and prediction on sequential images of a pendulum. **a)** $R^2$ for decoding of the pendulum angular velocity; the architectures achieve similar scores on inference, but aBFS outperforms the others on prediction (forecast). **b)** forecasted pendulum images.

nonlinear state-space models; the temporal structure of the posterior is imposed through structured inference networks, whereas we do so by parameterizing backward transitions using the prior. The structured variational autoencoder Johnson et al. (2016); Zhao & Linderman (2023) incorporates the prior by combining learned conjugate potentials with the prior, which is similar to the amortized mean parameter gradients of the expected log-likelihood that we consider; however, we don't restrict ourselves to priors (e.g. LDS) that make it easy to calculate the approximate posterior. In Karl et al. (2016) the inference model is forced to propagate samples through the prior so that gradients from the dynamics propagate through the expected log-likelihood term of the ELBO; aBFS allows this problem to be solved through masking similar to the semi-supervised mechanism in Zhao & Linderman (2023). In contrast to expectation propagation (EP) (Minka, 2001; Opper, 1999), our approach optimizes a global cost function. Those methods target probabilistic graphical models in a broad sense, while our work focuses on SSMs. Kamthe et al. (2022) also focus on structure preserving inference, but do so using EP and focus on Gaussian approximations.

## 5 EXPERIMENTS

**dBFS: examining convergence** We examine convergence of the dBFS described in Fig. 5. First, we consider a two dimensional linear dynamical system so that we can compare the variational posterior inferred to the true posterior; as an additional baseline we also consider a mean-field variational approximation given by $q(\mathbf{z}_{1:T}) = \prod \mathcal{N}(\mathbf{m}_t, \mathbf{P}_t)$. In Fig. 1a, we can see that dBFS recovers the true posterior in relatively few dual variable steps. We also consider a nonlinear dynamical system so that the true posterior cannot be found in closed form and approximate inference is necessary; as a baseline, we use the particle filter drawing 25k samples since this is low-dimensional (Douc et al., 2014). For this example, we draw trajectories from the Van der Pol oscillator project them to observation space and add Gaussian noise. In Fig. 1b, we see that dBFS approaches the solution inferred by the particle filter. From this experiment, we see that the ELBO achieved converges much more rapidly than the conditions for optimality of Eq. (24) and (23).

**aBFS: learning a pendulum** We consider a pendulum governed by Hamiltonian dynamics as in Botev et al. (2021). This Hamiltonian system is a good test bed for exploring the predictive capabilities of the learned dynamics; it is also particularly challenging – observations can be decoded with only angular position, however, predictions are only feasible if the model is able to learn the dynamics underlying the angular velocity. We compare aBFS to sequential VAE models of varying complexity. For these models, we factor the approximate posterior forward in time as $q(\mathbf{z}_{1:T} \mid \mathbf{y}_{1:T}) = \prod q(\mathbf{z}_t \mid \mathbf{z}_{t-1}, \mathbf{y}_{t:T})$. Furthermore, for the forward factorization we consider diffusing (D-VAE), linear(L-VAE) and nonlinear(N-VAE) parameterizations of the amortization network to model the distribution of $\mathbf{z}_{t+1} \mid \mathbf{z}_t$, i.e. we define the conditional means as,

$$\underset{\text{D-VAE}}{\mathbf{m}_{\phi_\text{D}}(\mathbf{z}_t, \mathbf{y}_{t:T}) = \alpha \mathbf{z}_t + \mathbf{h}_t} \quad \underset{\text{L-VAE}}{\mathbf{m}_{\phi_\text{L}}(\mathbf{z}_t, \mathbf{y}_{t:T}) = \mathbf{A}\mathbf{z}_t + \mathbf{h}_t} \quad \underset{\text{N-VAE}}{\mathbf{m}_{\phi_\text{N}}(\mathbf{z}_t, \mathbf{y}_{t:T}) = \mathbf{f}(\mathbf{z}_t) + \mathbf{h}_t} \quad (26)$$

where $\mathbf{h}_t$ is the output of some neural network (e.g. an RNN running backward in time) that encapsulates statistical information pertaining to $\mathbf{y}_{t:T}$. In Fig. 2b, we plot trajectories predicted from the learned dynamics for each model, showing that aBFS is able to perform long term forecasting. For D-VAE, there are a total of 107k parameters, 108k for L-VAE, and 113k for N-VAE, while for aBFS there are 117k parameters. In

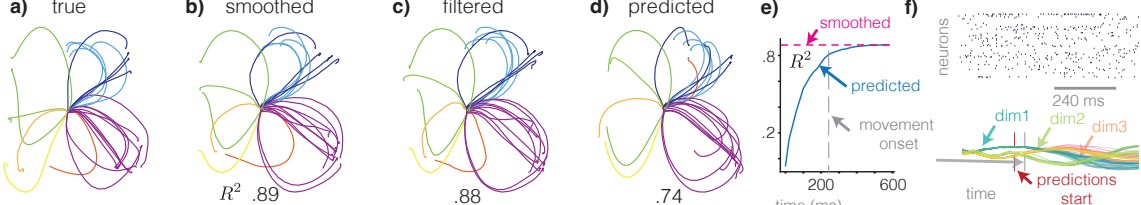

Figure 3: Modeling monkey's reaching. **a)** Actual reaches. Reaches linearly decoded from the inferred **b)** smoothed latent trajectories ($R^2 = 0.89$), **c)** causally filtered latent trajectories ($R^2 = 0.88$), **d)** latent trajectories predicted starting from an initial condition causally inferred before movement onset ($R^2 = 0.74$). Velocity decoding scores of similar methods include $0.891$ for iLQR-VAE (Schimel et al., 2021), $0.886$ for NDT (Ye & Pandarinath, 2021), $0.910$ for Auto-LFADS (Keshtkaran et al., 2022) (results taken from Schimel et al. (2021)). **e)** The $R^2$ value of the velocity decoded using predicted trajectories as a function of how far into the trial the latent state was filtered until it was only sampled from the autonomous dynamics; by the time of the movement onset, behavioral predictions using latent trajectories sampled through the dynamics are nearly on par with behavior decoded from the smoothed posterior statistics. **f)** (top) single-trial spike-train (bottom) sample trajectories of the top 3 latent dimensions.

Fig. 2a, we show the $R^2$ value for linear decoding of the angular velocity; aBFS is able to learn a dynamics model that retains information about angular velocity during prediction.

**Neurophysiological dataset: predicting neural population dynamics**  To evaluate the capability of extracting meaningful insights from real data, we apply aBFS to a neurophysiological dataset Churchland et al. (2012) that contains recordings from monkey motor cortex during a reaching task. This dataset has been used as a benchmark for the efficacy of latent variable models to neural data in previous studies (Pei et al., 2021). To accurately capture the variability of neural spike train observation, we choose the SSM such as a log-linear Poisson likelihood driven by a latent nonlinear dynamical system, i.e.,

$$p(\mathbf{y}_t \mid \mathbf{z}_t) = \text{Poisson}(\mathbf{y}_t \mid \exp(\mathbf{C}\mathbf{z}_t + \mathbf{b})) \qquad p_{\boldsymbol{\theta}}(\mathbf{z}_t \mid \mathbf{z}_{t-1}) = \mathcal{N}(\mathbf{z}_t \mid \mathbf{m}_{\boldsymbol{\theta}}(\mathbf{z}_{t-1}), \mathbf{Q}) \qquad (27)$$

We learned the underlying dynamical system and inferred the latent trajectories from neural spike trains. The aBFS faithfully captured the latent trajectories and dynamical system, which is reflected by the state of the art results of decoding and forecasting (Fig. 3). In Fig. 3 we also show what the decoded behavior could look in an online application if the aBFS learned local encoder were used without the backward encoder to approximate the filtering distribution. During training to disentangle the role of the local and backward encoder, we train on a convex combination of $\hat{\mathcal{L}}_B$ evaluated with the smoothing distribution, and the filtering distribution that can be inferred only with the local encoder.

## 6    CONCLUSION

In this work, we developed novel smoothing algorithms, dBFS and aBFS, for nonlinear SSMs with exponential family dynamics and arbitrary likelihoods. dBFS and aBFS take advantage of the proposed backward factorization and tractability afforded by the plug-in predictive and make it possible to use standard variational inference in a decoupled fashion or embrace the VAE framework and offload posterior computation onto learned inference networks. dBFS employs Lagrange multiplier that allows for parallelizable implementations well suited for modern computing architecture. aBFS allows for modern deep learning tools and learns expressive generative models in an end-to-end data driven fashion. In the future, we will be working on faster convergence of dBFS, and designing more sophisticated inference networks mentioned in Sec. 3.1. Future investigations will also attempt to rigorously quantify how detrimental use of plug-in approximations to simplify evaluation of the ELBO may be to the quality of the inferred posterior.

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

## A    NOMENCLATURE

| Symbol | Description |
|--------|------------|
| $q(\mathbf{z}_t)$ | smoothed variational approximation (with parameters $\boldsymbol{\lambda}_t$ / $\boldsymbol{\mu}_t$) |
| $\breve{q}(\mathbf{z}_t)$ | filtered variational approximation (with parameters $\breve{\boldsymbol{\lambda}}_t$ / $\breve{\boldsymbol{\mu}}_t$) |
| $\bar{q}(\mathbf{z}_t)$ | predictive variational approximation (with parameters $\bar{\boldsymbol{\lambda}}_t$ / $\bar{\boldsymbol{\mu}}_t$) |
| $p_{\boldsymbol{\theta}}(\mathbf{z}_t \mid \mathbf{z}_{t-1})$ | prior dynamics |
| $A(\cdot)$ | log-partition function |
| $A^*(\cdot)$ | log-partition function convex conjugate |
| $\boldsymbol{\lambda}$ | natural parameters |
| $\boldsymbol{\mu}$ | mean parameters, $\boldsymbol{\mu} = \mathbb{E}_q\left[t(\mathbf{z})\right]$ |
| $\mathbf{M}_t(\boldsymbol{\lambda}_t)$ | the Jacobian of the expected mean parameters, $\nabla_{\boldsymbol{\mu}_t}\mathbb{E}_{q_t}\left[\boldsymbol{\mu}_{\boldsymbol{\theta}}(\mathbf{z}_t)\right]$ |
| $\mathfrak{L}(\cdot)$ | function mapping $\boldsymbol{\mu}_t \mapsto \boldsymbol{\lambda}_t$ equivalent to $\nabla A^*(\boldsymbol{\mu})$ |
| $\mathfrak{D}(\cdot)$ | function mapping $\boldsymbol{\mu}_t \mapsto \bar{\boldsymbol{\mu}}_{t+1}$ equivalent to $\mathbb{E}_{q_t}\left[\boldsymbol{\mu}_{\boldsymbol{\theta}}(\mathbf{z}_t)\right]$ |
| $\mathfrak{F}(\cdot)$ | function mapping $\boldsymbol{\lambda}_t \mapsto \bar{\boldsymbol{\lambda}}_{t+1}$ |
| $\mathcal{I}(\boldsymbol{\lambda}_t)$ | Fisher information matrix |

Sometimes, we abbreviate distributions by dropping $\mathbf{z}_t$ as an argument, and just use the subscript $t$ with the appropriate decorator, e.g. $p_{t|t-1}$ for $p_{\boldsymbol{\theta}}(\mathbf{z}_t \mid \mathbf{z}_{t-1})$, or $q_t$ for $q(\mathbf{z}_t)$.

## B    LGSSM

A linear Gaussian state-space model (LGSSM) is defined as
$$p_{\boldsymbol{\theta}}(\mathbf{z}_{t+1} \mid \mathbf{z}_t) = \mathcal{N}(\mathbf{z}_{t+1} \mid \mathbf{A}\mathbf{z}_t, \mathbf{Q}) \qquad p(\mathbf{y}_t \mid \mathbf{z}_t) = \mathcal{N}(\mathbf{y}_t \mid \mathbf{C}\mathbf{z}_t, \mathbf{R}), \qquad (28)$$
The natural parameter mapping induced by the dynamics, in addition to the corresponding mean parameter mapping, are both affine in the sufficient statistics of the conditioning variable. This lets us transform the LGSSM dynamics into the representation of Eq. (3) by writing,
$$\boldsymbol{\lambda}_{\boldsymbol{\theta}}(\mathbf{z}_t) = \mathbf{F}t(\mathbf{z}_t) + \mathbf{f} \qquad \boldsymbol{\mu}_{\boldsymbol{\theta}}(\mathbf{z}_t) = \mathbf{G}t(\mathbf{z}_t) + \mathbf{g} \qquad (29)$$
where $t(\mathbf{z}_t) = \left(\mathbf{z}_t^\top \quad -\frac{1}{2}\mathrm{vec}(\mathbf{z}_t\mathbf{z}_t^\top)^\top\right)^\top$ and,
$$\mathbf{F} = \begin{pmatrix}\mathbf{Q}^{-1}\mathbf{A} & \mathbf{0} \\ \mathbf{0} & \mathbf{0}\end{pmatrix} \quad \mathbf{f} = \begin{pmatrix}\mathbf{0} \\ \mathbf{Q}^{-1}\end{pmatrix} \quad \mathbf{G} = \begin{pmatrix}\mathbf{A} & \mathbf{0} \\ \mathbf{0} & \mathbf{A}\otimes\mathbf{A}\end{pmatrix} \quad \mathbf{g} = -\frac{1}{2}\begin{pmatrix}\mathbf{0} \\ \mathbf{Q}\end{pmatrix}. \qquad (30)$$

## C    USEFUL FACTS

If $q$ is a minimal exponential family distribution, then, there exists a one-to-one mapping between its natural parameters, $\boldsymbol{\lambda}$, and its mean parameters, $\boldsymbol{\mu} := \mathbb{E}_q\left[t(\mathbf{z})\right]$ given by $\boldsymbol{\mu} = \nabla A(\boldsymbol{\lambda})$ (Wainwright & Jordan, 2008). We also have the following useful relation between the gradient with respect to mean parameters, $\boldsymbol{\mu}$, the gradient with respect to the natural parameters, $\boldsymbol{\lambda}$, and the Fisher information matrix, $\mathcal{I}(\boldsymbol{\lambda}) = \nabla^2 A(\boldsymbol{\lambda})$, (Khan & Lin, 2017)
$$\frac{\partial}{\partial\boldsymbol{\mu}} = \mathcal{I}(\boldsymbol{\lambda})^{-1}\frac{\partial}{\partial\boldsymbol{\lambda}} \qquad (31)$$
so that, $\nabla_{\boldsymbol{\mu}}\boldsymbol{\lambda} = \mathcal{I}^{-1}(\boldsymbol{\lambda})$.

If $q$ and $\bar{q}$ are both members of an exponential family of distributions, $\mathcal{Q}$, with natural parameters $\boldsymbol{\lambda}$ and $\bar{\boldsymbol{\lambda}}$ respectively, then the KL divergence has the following two useful expressions

$$\mathbb{D}_{\mathrm{KL}}(q\,||\,\bar{q}) = \boldsymbol{\mu}^\top(\boldsymbol{\lambda} - \bar{\boldsymbol{\lambda}}) - A(\boldsymbol{\lambda}) + A(\bar{\boldsymbol{\lambda}}) \tag{32}$$

$$= A^*(\boldsymbol{\mu}) + A(\bar{\boldsymbol{\lambda}}) - \boldsymbol{\mu}^\top\bar{\boldsymbol{\lambda}} \tag{33}$$

from which the gradients of the KL are,

$$\nabla_{\boldsymbol{\mu}}\mathbb{D}_{\mathrm{KL}}(q\,||\,\bar{q}) = \boldsymbol{\lambda} - \bar{\boldsymbol{\lambda}} \tag{34}$$

$$\nabla_{\bar{\boldsymbol{\mu}}}\mathbb{D}_{\mathrm{KL}}(q\,||\,\bar{q}) = \mathcal{I}(\bar{\boldsymbol{\lambda}})^{-1}(\bar{\boldsymbol{\mu}} - \boldsymbol{\mu}) \tag{35}$$

From facts about the cumulant generating function, (Seeger, 2005)

$$\begin{aligned}
&\log\left\{\mathbb{E}_{q(\mathbf{z};\boldsymbol{\eta})}\left[\exp\left(t(\mathbf{z})^\top\boldsymbol{\lambda}\right)\right]\right\} \\
&= \log\left\{\int h(\mathbf{z})\exp\left[t(\mathbf{z})^\top\boldsymbol{\eta} - A(\boldsymbol{\eta})\right]\exp\left[t(\mathbf{z})^\top\boldsymbol{\lambda}\right]\,\mathrm{d}\mathbf{z}\right\} \\
&= \log\left\{\int h(\mathbf{z})\exp\left[t(\mathbf{z})^\top\boldsymbol{\eta} - A(\boldsymbol{\eta}) + t(\mathbf{z})^\top\boldsymbol{\lambda}\right]\,\mathrm{d}\mathbf{z}\right\} \\
&= \log\left\{\int h(\mathbf{z})\exp\left[t(\mathbf{z})^\top(\boldsymbol{\eta} + \boldsymbol{\lambda}) - A(\boldsymbol{\eta})\right]\,\mathrm{d}\mathbf{z}\right\} \\
&= \log\left\{\underbrace{\int h(\mathbf{z})\exp\left[t(\mathbf{z})^\top(\boldsymbol{\eta} + \boldsymbol{\lambda}) - A(\boldsymbol{\eta} + \boldsymbol{\lambda})\right]\,\mathrm{d}\mathbf{z}}_{=1}\right\} + A(\boldsymbol{\eta} + \boldsymbol{\lambda}) - A(\boldsymbol{\eta}) \\
&= A(\boldsymbol{\eta} + \boldsymbol{\lambda}) - A(\boldsymbol{\eta})
\end{aligned} \tag{36}$$

## D  MINIMIZER OF THE CONDITIONAL FORWARD KL

The optimal mean variational parameters under the forward KL objective satisfy

$$\bar{\boldsymbol{\mu}}^* = \mathbb{E}_{\bar{p}(\mathbf{z}_{t+1})}\left[t(\mathbf{z}_{t+1})\right] \tag{37}$$

where

$$\bar{p}(\mathbf{z}_{t+1}) = \mathbb{E}_{q(\mathbf{z}_t)}\left[p_{\boldsymbol{\theta}}(\mathbf{z}_{t+1}\mid\mathbf{z}_t)\right] \tag{38}$$

which we can try to simplify as

$$\bar{\boldsymbol{\mu}}^* = \int t(\mathbf{z}_{t+1})\,\mathbb{E}_{q(\mathbf{z}_t)}\left[p_{\boldsymbol{\theta}}(\mathbf{z}_{t+1}\mid\mathbf{z}_t)\right]\mathrm{d}\mathbf{z}_{t+1} \tag{39}$$

$$= \int t(\mathbf{z}_{t+1}) \tag{40}$$

$$\times\left[\int h(\mathbf{z}_t)h(\mathbf{z}_{t+1})\exp\left(t(\mathbf{z}_{t+1})^\top\boldsymbol{\lambda}_{\boldsymbol{\theta}}(\mathbf{z}_t) + t(\mathbf{z}_t)^\top\boldsymbol{\lambda} - A(\boldsymbol{\lambda}) - A(\boldsymbol{\lambda}_{\boldsymbol{\theta}}(\mathbf{z}_t))\right)\mathrm{d}\mathbf{z}_t\right]\mathrm{d}\mathbf{z}_{t+1}$$

$$= \int h(\mathbf{z}_t)\exp\left(t(\mathbf{z}_t)^\top\boldsymbol{\lambda} - A(\boldsymbol{\lambda})\right) \tag{41}$$

$$\times\left[\int t(\mathbf{z}_{t+1})h(\mathbf{z}_{t+1})\exp\left(t(\mathbf{z}_{t+1})^\top\boldsymbol{\lambda}_{\boldsymbol{\theta}}(\mathbf{z}_t) - A(\boldsymbol{\lambda}_{\boldsymbol{\theta}}(\mathbf{z}_t))\right)\mathrm{d}\mathbf{z}_{t+1}\right]\mathrm{d}\mathbf{z}_t$$

$$= \mathbb{E}_{q(\mathbf{z}_t)}\left[\nabla A(\boldsymbol{\lambda}_{\boldsymbol{\theta}}(\mathbf{z}_t))\right] \tag{42}$$

$$= \mathbb{E}_{q(\mathbf{z}_t)}\left[\boldsymbol{\mu}\left(\boldsymbol{\lambda}_{\boldsymbol{\theta}}(\mathbf{z}_t)\right)\right] \tag{43}$$

$$= \mathbb{E}_{q(\mathbf{z}_t)}\left[\boldsymbol{\mu}_{\boldsymbol{\theta}}(\mathbf{z}_t)\right] \tag{44}$$

# E    STATIONARY CONDITIONS FOR THE UNCONSTRAINED ELBO

The Lagrangian of the ELBO is given by,

$$\widehat{\mathcal{L}}_{B,U}(\boldsymbol{\lambda}_{1:T}, \bar{\boldsymbol{\mu}}_{2:T}, \boldsymbol{\nu}_{2:T}) = \sum_{t=1}^{T} \mathbb{E}_{q_t} \left[ \log p(\mathbf{y}_t \mid \mathbf{z}_t) \right] - \mathbb{D}_{\text{KL}}(q_t || \bar{q}_t) - \boldsymbol{\nu}_t^{\top} \left( \bar{\boldsymbol{\mu}}_t - \mathbb{E}_{q_{t-1}} \left[ \boldsymbol{\mu_\theta}(\mathbf{z}_{t-1}) \right] \right) \quad (45)$$

with $\bar{q}(\mathbf{z}_1) := p_{\boldsymbol{\theta}}(\mathbf{z}_1)$. Taking gradients with respect to the mean parameters, $\boldsymbol{\mu}_t$ and $\bar{\boldsymbol{\mu}}_t$, we have that the stationary conditions are

$$\nabla_{\boldsymbol{\mu}_t} \mathcal{L} = \tilde{\boldsymbol{\lambda}}(\mathbf{y}_t, \boldsymbol{\lambda}_t) - (\boldsymbol{\lambda}_t - \bar{\boldsymbol{\lambda}}_t) + [\mathbf{M}_t(\boldsymbol{\lambda}_t)]^{\top} \boldsymbol{\nu}_{t+1} = 0 \qquad t = 1, \ldots, T-1 \quad (46)$$

$$\nabla_{\boldsymbol{\mu}_T} \mathcal{L} = \tilde{\boldsymbol{\lambda}}_T - \boldsymbol{\lambda}_T + \bar{\boldsymbol{\lambda}}_T = 0 \quad (47)$$

$$\nabla_{\bar{\boldsymbol{\mu}}_t} \mathcal{L} = \mathcal{I}(\bar{\boldsymbol{\lambda}}_t)^{-1}(\boldsymbol{\mu}_t - \bar{\boldsymbol{\mu}}_t) - \boldsymbol{\nu}_t = 0 \quad (48)$$

which mean at a stationary point of the augmented ELBO the dynamics for $\boldsymbol{\lambda}_t$ obey the following recursion forward in time,

$$\boldsymbol{\lambda}_t = \mathfrak{F}(\boldsymbol{\lambda}_{t-1}) + [\mathbf{M}_t(\boldsymbol{\lambda}_t)]^{\top} \boldsymbol{\nu}_{t+1} + \tilde{\boldsymbol{\lambda}}(\mathbf{y}_t, \boldsymbol{\lambda}_t) \quad (49)$$

and the Lagrange multipliers are the Fisher scaled difference of mean parameters

$$\boldsymbol{\nu}_t = \mathcal{I}(\bar{\boldsymbol{\lambda}}_t)^{-1}(\boldsymbol{\mu}_t - \bar{\boldsymbol{\mu}}_t) \quad (50)$$

and satisfy a backward recursion given by,

$$\boldsymbol{\nu}_t = \mathcal{I}(\bar{\boldsymbol{\lambda}}_t)^{-1} \left( \mathfrak{L}^{-1} \left( \bar{\boldsymbol{\lambda}}_t + [\mathbf{M}_t(\boldsymbol{\lambda}_t)]^{\top} \boldsymbol{\nu}_{t+1} + \tilde{\boldsymbol{\lambda}}(\mathbf{y}_t, \boldsymbol{\lambda}_t) \right) - \bar{\boldsymbol{\mu}}_t \right) \quad (51)$$

# F    VARIATIONAL UPDATES (INNER OPTIMIZATION)

## F.1    $\widehat{\mathcal{L}}_{B,U}(q)$

One step of natural gradient ascent updates $\boldsymbol{\lambda}_{1:T}$ and $\bar{\boldsymbol{\lambda}}_{1:T}$ according to,

$$\boldsymbol{\lambda}_t^{(k+1)} = \boldsymbol{\lambda}_t^{(k)} + \alpha \left( \tilde{\boldsymbol{\lambda}} \left( \mathbf{y}_t, \boldsymbol{\lambda}_t^{(k)} \right) - \left( \boldsymbol{\lambda}_t^{(k)} - \bar{\boldsymbol{\lambda}}_t^{(k)} \right) + \left[ \mathbf{M}_t \left( \boldsymbol{\lambda}_t^{(k)} \right) \right]^{\top} \boldsymbol{\nu}_{t+1} \right) \quad (52)$$

$$= (1-\alpha)\boldsymbol{\lambda}_t^{(k)} + \alpha \left( \bar{\boldsymbol{\lambda}}_t^{(k)} + \tilde{\boldsymbol{\lambda}} \left( \mathbf{y}_t, \boldsymbol{\lambda}_t^{(k)} \right) + \left[ \mathbf{M}_t \left( \boldsymbol{\lambda}_t^{(k)} \right) \right]^{\top} \boldsymbol{\nu}_{t+1} \right) \quad (53)$$

$$\bar{\boldsymbol{\lambda}}_t^{(k+1)} = \bar{\boldsymbol{\lambda}}_t^{(k)} + \alpha \left( \mathcal{I}(\bar{\boldsymbol{\lambda}}_t^{(k)})^{-1}(\boldsymbol{\mu}_t^{(k)} - \bar{\boldsymbol{\mu}}_t^{(k)}) - \boldsymbol{\nu}_t \right) \quad (54)$$

## F.2    MEAN-FIELD

For a mean-field factorization, the ELBO reduces to

$$\mathcal{L}_{MF}(q) = \sum \mathbb{E}_{q_t} \left[ \log p(\mathbf{y}_t \mid \mathbf{z}_t) \right] - \mathbb{E}_{q_{t-1}} \left[ \mathbb{D}_{\text{KL}} \left( q_t || p_{t|t-1} \right) \right] \quad (55)$$

and the natural gradient updates are

$$\boldsymbol{\lambda}_t^{(k+1)} = (1-\alpha)\boldsymbol{\lambda}_t^{(k)} + \alpha \left( \tilde{\boldsymbol{\lambda}}_t + \mathbb{E}_{q_{t-1}} \left[ \boldsymbol{\lambda_\theta}(\mathbf{z}_{t-1}) \right] \right) \quad (56)$$

## F.3    $\mathcal{L}_B(q)$

For the regular backward factorized ELBO,

$$\mathcal{L}_B(q) = \sum \mathbb{E}_{q_t} \left[ \log p(\mathbf{y}_t \mid \mathbf{z}_t) \right] - \mathbb{D}_{\text{KL}}(q_t || \bar{q}_t) \quad (57)$$

we have that the natural gradient updates are given by

$$\boldsymbol{\lambda}_t^{(k+1)} = \boldsymbol{\lambda}_t^{(k)} + \alpha \nabla_{\boldsymbol{\mu}_t} \mathcal{L} \tag{58}$$

$$= \boldsymbol{\lambda}_t^{(k)} + \alpha \left( \tilde{\boldsymbol{\lambda}}_t - \nabla_{\boldsymbol{\mu}_t} \mathbb{D}_{\mathrm{KL}}(q_t || \bar{q}_t) - \nabla_{\boldsymbol{\mu}_t} \mathbb{D}_{\mathrm{KL}}(q_{t+1} || \bar{q}_{t+1}) \right) \tag{59}$$

$$= \boldsymbol{\lambda}_t^{(k)} + \alpha \left( \tilde{\boldsymbol{\lambda}}_t - (\boldsymbol{\lambda}_t - \bar{\boldsymbol{\lambda}}_t) - [\nabla_{\boldsymbol{\mu}_t} \bar{\boldsymbol{\mu}}_{t+1}] \mathcal{I}(\bar{\boldsymbol{\lambda}}_{t+1})^{-1} (\bar{\boldsymbol{\mu}}_{t+1} - \boldsymbol{\mu}_{t+1}) \right) \tag{60}$$

$$= (1 - \alpha)\boldsymbol{\lambda}_t^{(k)} + \alpha \left( \tilde{\boldsymbol{\lambda}}_t + \bar{\boldsymbol{\lambda}}_t^{(k)} + \mathbf{M}_t(\boldsymbol{\lambda}_t) \mathcal{I}(\bar{\boldsymbol{\lambda}}_{t+1})^{-1} (\boldsymbol{\mu}_{t+1} - \bar{\boldsymbol{\mu}}_{t+1}) \right) \tag{61}$$

## G  EXPANDING THE SMOOTHING OBJECTIVE

To simplify the smoothing objective, we have to consider the term

$$\log \int \frac{p(\mathbf{z}_{t+1} \mid \mathbf{z}_t) q(\mathbf{z}_{t+1})}{\bar{q}(\mathbf{z}_{t+1})} \, d\mathbf{z}_{t+1} \tag{62}$$

which from Eq. (36), can be rewritten as

$$\log \left( \mathbb{E}_{q(\mathbf{z}_{t+1})} \left[ \exp \left( t(\mathbf{z}_{t+1})^\top (\boldsymbol{\lambda}_{\boldsymbol{\theta}}(\mathbf{z}_t) - \bar{\boldsymbol{\lambda}}) - A(\boldsymbol{\lambda}_{\boldsymbol{\theta}}(\mathbf{z}_t)) + A(\bar{\boldsymbol{\lambda}}) \right) \right] \right) \tag{63}$$

$$= A(\bar{\boldsymbol{\lambda}}) - A(\boldsymbol{\lambda}_{\boldsymbol{\theta}}(\mathbf{z}_t)) + \log \left( \mathbb{E}_{q(\mathbf{z}_{t+1})} \left[ \exp \left( t(\mathbf{z}_{t+1})^\top (\boldsymbol{\lambda}_{\boldsymbol{\theta}}(\mathbf{z}_t) - \bar{\boldsymbol{\lambda}}) \right) \right] \right) \tag{64}$$

$$= A(\bar{\boldsymbol{\lambda}}) - A(\boldsymbol{\lambda}_{\boldsymbol{\theta}}(\mathbf{z}_t)) + A(\boldsymbol{\lambda}_{\boldsymbol{\theta}}(\mathbf{z}_t) - \bar{\boldsymbol{\lambda}} + \boldsymbol{\lambda}) - A(\boldsymbol{\lambda}) \tag{65}$$

## H  EXPERIMENTAL DETAILS

### H.1  ABFS / DBFS ALGORITHMS

### H.2  LDS AND VAN DER POL

We generate trajectories from an two dimensional LDS with rotational dynamics, $p(\mathbf{z}_t \mid \mathbf{z}_{t-1}) = \mathcal{N}(\mathbf{A}\mathbf{z}_{t-1}, \mathbf{Q})$, where we set $\mathbf{Q} = 0.25^2\mathbf{I}$. The likelihood is set to be $p(\mathbf{y}_t \mid \mathbf{z}_t) = \mathcal{N}(\mathbf{C}\mathbf{z}_t, \mathbf{R})$, where $\mathbf{R} = 0.5^2\mathbf{I}$. We use a learning rate of $0.05$ for the dual variable, $0.01$ for the inner natural gradient steps. dBFS is run for 100 dual variable gradient steps.

For the Van der Pol system, the generative model is

$$\mathbf{z}_{t+1,1} = \mathbf{z}_{t,1} + \frac{1}{\tau_1} \Delta \mathbf{z}_{t,2} + \sigma\epsilon \tag{66}$$

$$\mathbf{z}_{t+1,2} = \mathbf{z}_{t,2} + \frac{1}{\tau_2} \Delta(\gamma(1 - \mathbf{z}_{t,1})^2 \mathbf{z}_{t,2} - \mathbf{z}_{t,1}) + \sigma\epsilon \tag{67}$$

$$\mathbf{y}_t \mid \mathbf{z}_t \sim \mathcal{N}(\mathbf{y}_t \mid \mathbf{C}\mathbf{z}_t, \mathbf{R}) \tag{68}$$

where $\epsilon$ is white Gaussian noise. We set $\gamma = 1.5$, $\tau_1 = \tau_2 = 0.1$, $\Delta = 0.005$, $\sigma = 0.05$, and $\mathbf{R} = 0.1^2\mathbf{I}$, and generate 500 trials of length 200 so that L-VAE can learn a good inference network. For the dual variable updates we use a learning rate of $0.05$ and the inner optimization uses a step size of $0.01$ for the natural gradients.

### H.3  PENDULUM

For generating the pendulum dataset, we take advantage of the code available from Missel (2022). We generate 500 trajectories of length 75 for the training set and corrupt the observations with Gaussian noise with $\sigma = 0.01$. For all of the inference models and dynamics models, we use Adam with a learning rate of $0.0005$ (Kingma & Ba, 2014), and train the models for 150 epochs. In order to focus on the algorithms capability of learning a good dynamics model, we pre-train the neural network used for the observation model.

**a)** `dBF_smoothing` $(\mathbf{y}_{1:T})$

`elbo_argmax` $(\mathbf{y}_{1:T}, \boldsymbol{\lambda}_{1:T}, \bar{\boldsymbol{\lambda}}_{2:T}, \boldsymbol{\nu}_{2:T})$

`until convergence:`

  # solve primal

  $\boldsymbol{\lambda}_{1:T}, \bar{\boldsymbol{\lambda}}_{2:T} = $ `elbo_argmax` $\left(\mathbf{y}_{1:T}, \boldsymbol{\lambda}_{1:T}, \bar{\boldsymbol{\lambda}}_{2:T}, \boldsymbol{\nu}_{2:T}^{(\ell)}\right)$

  # dual update (for all t, in parallel)

  $\boldsymbol{\nu}_t^{(\ell+1)} = \boldsymbol{\nu}_t^{(\ell)} + \gamma \left( \bar{\boldsymbol{\mu}}_t - \mathbb{E}_{q_{t-1}}\left[ \boldsymbol{\mu}_{\boldsymbol{\theta}}(\mathbf{z}_{t-1}) \right] \right)$

    `until convergence:`

      # smoothed parameter update

      $\boldsymbol{\lambda}_t^{(k+1)} = (1-\alpha)\boldsymbol{\lambda}_t^{(k)} + \alpha(\bar{\boldsymbol{\lambda}}_t^{(k)} + \tilde{\boldsymbol{\lambda}}_t + \mathbf{M}_t^\top \boldsymbol{\nu}_{t+1})$

      # predictive parameter update

      $\bar{\boldsymbol{\lambda}}_t^{(k+1)} = \bar{\boldsymbol{\lambda}}_t^{(k)} + \alpha \left( \mathcal{I}(\bar{\boldsymbol{\lambda}}_t^{(k)})^{-1}(\boldsymbol{\mu}_t^{(k)} - \bar{\boldsymbol{\mu}}_t^{(k)}) - \boldsymbol{\nu}_t \right)$

**b)** `aBF_smoothing` $(\mathbf{y}_{1:T})$

`sample_nat_params` $(\mathbf{y}_{1:T})$

`until convergence:`

  $\boldsymbol{\lambda}_{1:T}, \bar{\boldsymbol{\lambda}}_{2:T} = $ `sample_nat_params` $(\mathbf{y}_{1:T})$

  $\widehat{\mathcal{L}}_B = \sum \mathbb{E}_{q_t} \left[ \log p(\mathbf{y}_t \mid \mathbf{z}_t) \right] - \mathbb{D}_{\mathrm{KL}}(q_t || \bar{q}_t)$

  $\boldsymbol{\phi} = \boldsymbol{\phi} + \beta \nabla_{\boldsymbol{\phi}} \widehat{\mathcal{L}}_B$

    `for t in [1:T]:`

      # dynamics        # inference networks

      $\bar{\boldsymbol{\mu}}_t = \mathbb{E}_{q_{t-1}}\left[ \boldsymbol{\mu}_{\boldsymbol{\theta}}(\mathbf{z}_{t-1}) \right]$   $\mathbf{u}_t = \mathrm{NN}(\mathbf{y}_{t+1:T}; \boldsymbol{\phi}_1)$

      $\bar{\boldsymbol{\lambda}}_t = \mathfrak{L}(\bar{\boldsymbol{\mu}}_t)$         $\tilde{\boldsymbol{\lambda}}_t = \mathrm{NN}(\mathbf{y}_t; \boldsymbol{\phi}_2)$

      $\boldsymbol{\lambda}_t = \bar{\boldsymbol{\lambda}}_t + \tilde{\boldsymbol{\lambda}}_t + \mathbf{u}_t$

    `return` $\boldsymbol{\lambda}_{1:T}, \bar{\boldsymbol{\lambda}}_{2:T}$

Figure 4: **a)** dBFS: we decouple the dependence of neighboring natural parameters so they can be optimized locally, only exchanging information through the Lagrange multipliers. **b)** aBFS: in the inner loop, we sample natural parameters forward using Eq. (25) with the prior to compute predictive parameters and inference networks for the others; in the outer loop we update the inference networks.

We use the PyTorch (Paszke et al., 2019) GRUCell to parameterize the model of the dynamics. For aBFS we parameterize $\tilde{\boldsymbol{\lambda}}_{\boldsymbol{\phi}}$ using a single layer 256 hidden unit MLP with Swish nonlinearity (Ramachandran et al., 2017). For all of the RNNs running backward in time, we parameterize them as single layer, 128 hidden unit GRUs. Like aBFS, for N-VAE, we parameterize the conditional mean mapping with a 256 hidden unit MLP with Swish nonlinearity. For the traditional VAEs, we mask points by randomly selecting time points while sampling trajectories to construct their posterior statistics by passing the previous time-step mean parameters through the dynamics rather than the inference network.

