# OpenReview forum: "Smoothing for exponential family dynamical systems"
_ICLR.cc/2024/Conference — Submitted to ICLR 2024_

### Official Review · Reviewer_CGvB · 2023-10-31

**Soundness:** 3 good
**Presentation:** 3 good
**Contribution:** 2 fair
**Rating:** 5
**Confidence:** 4

**Summary:**

The paper develops three methods for smoothing in state-space models (SSMs). The idea is to assume SSMs that are non-linear and avoid assumptions like Gaussianity when using variational inference. The driving idea is thus to preserve the temporal structure in the variational proposal. This seems to lead to what is called "exponential family dynamical systems", that is, a double-looped (forward and backward) chain of Markovian conditionals.

**Strengths:**

Having carefully checked the exponential family derivations, the parameterization, as well as the derived ELBOs, I feel that likely they are correct and well-founded on previous related work. The use of exponential families in this context, and particularly to build the factorization into Markovian conditionals is definitely a strength. The work itself is clear and concise in the details, also mentioning limitations and reasoning on why certain decisions are taken.

**Weaknesses:**

To me, the paper has two main points of weaknesses:

[w1] — the work is in general concise and thorough, but written in a way that the smoothing idea is kind of lost. Particularly, technical details jump in to solve issues of previous technical details (derivations begin at the beginning of pp. 2 and finish at the end of pp. 7). In that way, the paper loses quite a lot of space, and story on the general smoothing idea that authors want to solve (and in which way they want to solve it). This is orthogonal to the fact that the technical details and decisions are elegant and well-constructed

[w2] — the second concern to me is the limited results. Having derived long technical details, with the approximated ELBO, the Lagrangian-based optimization and the additional algorithm, the manuscript should at least provide results proportional to the technical development. In my opinion, the evaluation of the model is somehow short (learning of two synthetic systems (pendulum and chaotic scenario) plus analysis on convergence). In this sense, I would have liked to see experiments proving the feasibility of the model with long time data or similar. (Mainly bc it is indicated that the proposed ELBO delivers a computational cost equivalent to mean-field approximations without such crude factorizations).

**Questions:**

Due to the accidental mistake that I made with the previous review, and the lack of time that I unfortunately gave to the authors on this rebuttal, I will focus my questions only on one point:

I'm sort of concerned with the effect of the plug-in predictive distribution in Eq. 13. In particular, I sell the decisions taken and I liked a lot the sort of factorization introduced to build the new exponential-family-based ELBO with such structure. However, re-introducing an approximation of this predictive quantity in the ELBO to obtain another approximation of the target objective makes me wonder if the ELBO is 1) still tight wrt. the marginal likelihood, 2) if it is still a lower bound, and 3) if it has drawbacks or additional problems. In that direction, details are not clear to me and I would love to hear back if there is some time still available.

---

> ### Author Response · Authors · 2023-11-13
>
> Dear reviewer,
>
> We would like to kindly ask if the review uploaded isn't for a different paper than this one.  If that is the case, we hope to receive your review so that we could address any concerns you may have.
>
> Best regards,
> the authors

---

> ### Comment · Reviewer_CGvB · 2023-11-21
> **Apologies and Review update**
>
> I sincerely want to apologize to the authors (AC, PCs, and the rest of the reviewers) for my mistake on the initial review, which I accidentally swapped with another submission with a "similar" ID number. I also want to apologize to the authors for receiving a confusing review and if this has caused a bad impression of the reviewing process of ICLR, which I remark is still rigorous and aware of the mistake I accidentally committed. Last but not least, I want to remark my positive consideration of the work which is thorough and of high technical value, even if I remarked some points in my review.
>
> The correct review of this paper is now visible again and in its correct form.

---

> ### Author Response · Authors · 2023-11-23
>
> Dear reviewer,
>
> Thank you for taking the time to provide feedback on the manuscript.  Motivated by concerns of the limited results, we have included an additional **real-data experiment** in order to demonstrate the efficacy of the ideas presented applied to learning a dynamical system from neural data.  Motivated by other reviewer suggestions, we have also removed the vBF section in favor of using the additional space to add more details and have a more cohesive story.  Additionally, we respond to your questions below
>
> 1) In this case, the ELBO would not be tight in the general case; but we find it reassuring that the plugin predictive approximation is exact for the linear Gaussian state-space model, and we have tried to include that in the text to aid in building intuition. We have evaluated the tightness empirically in Fig.1. And considering the Kalman smoother and particle filter marginal likelihoods for the example linear and nonlinear dynamical systems, we can see our ELBOs (purple dashed lines) are close to the marginals.
>
> 2) This is a good point. The objective ELBO defined by Eq.14 is still a lower bound of some marginal likelihood by its form. However, it corresponds to a prior different from (approximating) the original one defined by the generative model. Many other approaches in fact have to use approximation too, and thus implicitly introduced a new prior/marginal as well.
>
> 3) Theoretically it leads to an ELBO of a different marginal as we mentioned in 2) and in turn might increase or decrease the variational gap. Empirically, we have not found this to be a problem in the experiments performed.
>
> Thank you again for your feedback.  We feel that the additional experiment and approximation details motivated by your questions have helped increase the quality of our manuscript greatly.  We hope that you could consider adjusting your score if you find it warranted given the extra improvements.
>
> Thanks for your time,
>
> The authors

---

### Official Review · Reviewer_pQjy · 2023-10-31

**Soundness:** 3 good
**Presentation:** 3 good
**Contribution:** 3 good
**Rating:** 6
**Confidence:** 2

**Summary:**

This paper investigates a new factorization of the ELBO for sequence models.  This factorization uses a structured backwards-factored proposal that leverages the forward dynamics.  This choice, while appealing for analysis purposes, presents a number of inference difficulties.  The contributions of the paper is then a set of methods for performing approximate inference within the variational framework.  The method is tested on some small problems.

**Note** It is important for me to note that while I understand most of the paper, this is right on the edge of what I am comfortable reviewing.  There are parts of the paper that I do not understand, not because I think they are wrong, but because I have struggled to follow the thread in places.

**Strengths:**

The work dovetails nicely with some recent work studying how structuring variational posteriors can lead to better inferences, but is awkward to learn.  The proposed method is a clever way of working around these problems, and expands the families of models that can be analyzed.  I think the method could be experimentally very useful, and engender follow-up work.

The paper itself is well written, is self-contained, and has a good degree of purely pedagogical merit.  The authors are to be commended for their attention to detail, and the thought that has clearly been dedicated to structuring the paper.

**Weaknesses:**

Firstly, I believe the core work is at the requisite level for publication.  The work is sound, and to the best of my knowledge, is novel.  However, I do have numerous questions for the authors as there are aspects of the work I do not understand.  These are listed below.

My main criticisms of the work breaks down into two parts: the empirical evaluation, and the overall clarity.

**W.1.:  Experimental evaluation**:  I think the experimental evaluation of the proposed methods is weak.  The problem with simultaneously proposing several methods is that you evaluate each method individually less.  Furthermore, the description of the experiments is comfortably the worst prose in the submission.  For instance, I have pretty much no idea what is being studied in “dBFS: convergence”.  I have outlined my comments on this below.  Evaluation of a single variant on a Van De Pol oscillator, and evaluation of some methods on a chaotic RNN and on a simple pendulum are not exactly compelling applications of a method.  Is the autoencoding method better than the Lagrange method?  Does one have more parameters, hyperparameters, more expensive etc.  There is no actual evaluation of the relative merits of the methods presented.

There are also no compelling baselines or applications.  For instance, the original SVAE was applied to analyze mouse data.  I would also like to see comparison to other VAE variants and inference techniques – VRNNs and DKFs immediately spring to mind.  I think there are also entire families of inference techniques that you could discuss, e.g. VIFLE and SIXO [Kim et al, 2020; Lawson et al 2022], that are untouched.  These are just examples, and need not be quantitative comparisons, but since you are proposing leveraging backwards information or factorizations, this is exactly what VIFLE and SIXO do.  In general, the linking into prior art is incredibly weak.

**W.2.:  Clarity**:  My main concern is not so much in the actual writing of the paper, but the clarity of the message.  I think the authors have tried to pile so many ideas and derivations into a single paper that the core message is actually obfuscated.

I’ll try and give an example.  If the Lagrange method is universally better than the autoencoding method (or vice versa) then omit the weaker method from the main text.  If there are conditions where one is better/preferable than the other, then there should be an explicit experiment tackling this, so that the reader can understand the difference.

Beyond this, I am not sure _why_ I should be interested in this method.  I have made some of these questions explicit in the Questions section.  It is stated early on that forwards and backwards factorizations have different analysis properties – this is then not mentioned throughout the paper.  It is also stated that a certain factorization underuses dynamics and is inefficient – this is not shown anywhere in the paper, but seems like a fairly fundamental justification for the method.  There are no experimental results that help justify the complexity or show that this method instantly overcomes some hurdles faced by previous methods.

There are no diagrams outlining the various steps or architectures;  tables compactly contrasting the methods or their free variables;  or summary paragraphs summarizing what was introduced.  All these are tools to make the paper easier to understand.  And easier to understand papers ultimately have a greater reach and impact!

There are also no fail cases or real limitations of the method stated – which for a method that doesn’t actually optimize a true ELBO and has a number of approximations is definitely required.

I think this paper would _greatly_ benefit from thinning out the content.  I appreciate the authors not trying to slice the contributions too thinly, but (again as an example) I query if the vBF connection really fits with this paper.  I would almost rather that be removed, to make more room for a more thorough and focused discussion and comparison of the dBFS and aBFS algorithms.  There would also be more space for a more thorough (quantitative or qualitative) comparison to existing works, and where this work improves over those methods.


## Minor comments:
- a. Throughout the paper, there are parameters ($\nu, \lambda, \ldots$) in equations or objectives that do not show up in the right hand side (e.g. $\lambda$ and $\theta$ in (17-18)).  It would be much better if the authors consistently and explicitly indicated which terms are indexed by which parameters.
- b. Tables and figures should be floated to the top or bottom of the page, as opposed to inlined in text (c.f. Figure 3).
- c. I don’t feel like Figure 1 actually helps that much.  I would rather it be written out as a proper pseudocode algorithm block that I can follow, as opposed to restating snippets of math that appear elsewhere.  (It’s also a huge figure that could be made _much_ smaller.  Side by side algorithms?)
- d. It would be nice if somewhere there was a table outlining the constraints on the model family that can be analyzed.
- e. I have completely lost the thread of how the methods interact with each other by the time we get to “vBF: smoothing chaotic dynamics”, and i’m not really sure how to read the results from the table.


## Comments on “dBFS convergence”:
- vBFS isn’t defined in the text.
- Why is Figure 2b studying an L-VAE?  What has that got to do with dBFS?
- (20) and (21) are a condition for optimality, and so how can you say the method converges _faster_?
- There is no other line to compare to on the LHS panel in each figure.
- Why are the particle filters bound so low?  PFs are asymptotically tight.  You should run it with a range of particles to see how many particles are required to get a tight bound.
- The middle panel is practically unexplained – is it good that the $\lambda$s are approaching zero?
- Colors and line styles are clashing.  Is the histogram over dBFS, L-VAE or mean field, or is it over 1, 10 and 100 steps.  Pick some different colors, or, give each subfigure its own legend.
- Is “no. dual steps” the same as the colored 1, 10, and 100?
- I don’t even really know what the objective of this experiment is.  I think it is showing/sanity-checking that variational inference works?  That is sort of to be expected.  But with nothing to compare it to, it isn’t telling me a great deal.


**Summary** I think this paper shows great promise.  The core of the work is clearly publication-standard, and the authors have tried hard to make it digestible.  Unfortunately right now I believe their efforts are slightly misguided, and have resulted in a paper that doesn’t quite hit the home run that it is so close to.  I would be happy to accept a paper where the vBF content is removed, and replaced with a more thorough discussion and evaluation of the core methods.  I have tried to be thorough in my questions below, and if answered, I am happy to upgrade my score.

**Questions:**

**Q.1.**:  Why does L have different parameters in (10) vs (11)? ($\lambda$ vs $q$)

**Q.2.**:    Just below (14), you point out that as a result of the approximations it is no longer a valid ELBO.  This seems bad (although not terminal), and I was surprised this issue is not referenced or addressed elsewhere.  If this is being optimized end-to-end, and the objective is not a true bound, then you cannot make guarantees on what you are optimizing.  Can the authors comment on this.

**Q.3.**:   At the bottom of page 4, you say the following:

```The backward factorization will make gaining insight into the structure of the variational posterior analytically easier; whereas $L_F(q)$ is easier to use in an amortized inference setting where trajectories from the approximate posterior are drawn and used to form a stochastic approximation of the bound (Krishnan et al., 2016; Karl et al., 2016)```.

Again, this is not obvious to me.  Nowhere else in the paper is the interpretability of the ELBO considered.  You also assert that $L_F$ is easier to evaluate compared to $L_B$, but then later on, you evaluate $L_B$, so I don't understand what the difference is.

**Q.4.**:  This method be applied to any model with exponential family potentials?  Can one mix and match potential types?  I’m thinking something like a Poisson-HMM would be a nice demonstrative example of this, and one where there are many readily available baselines you could compare to.

**Q.5.**:  (Following on from Q.4.)  Are the D-VAE, L-VAE and N-VAE all different model classes, or proposal classes?  It would be interesting to compare the inference performance of some different models on some different applications.  For instance, with sufficiently small discretization steps, an extended Kalman filter would solve the Van Der Pol oscillator problem.  How does the inference performance of both methods break down as that timestep grows larger?  Have you tested how close to an exponential family the empirical distributions are, and whether your inference objective learns models that have “better” approximate posteriors?

**Q.6.**:  Can the authors clarify why natural gradient steps are preferable?  What would happen if you didn’t use NG, and which other methods can use NG?

**Q.7.**: How computationally expensive is each method?  You say that the Lagrange step is parallelizable, but the autoencoding variant almost certainly isn’t for sequence data (unless you use something like S4 as the RNN).

**Q.8.**: I think I understand the constrained optimization step, but if the authors could clarify:  Instead of optimizing all the parameters freely, you switch to an EM-like alternating optimization, where the M-step looks like a constrained optimization.  This allows you to decouple the timesteps by essentially treating some of the summary statistics as fixed.  Does this coordinate-wise method have any drawbacks?

**Q.9.**: This is a relatively core question, but can the authors concretely re-state _why_ we should consider a backward factorization of the proposal?  Specifically which step cannot be achieved if we use a forward factorization?  And where does each method break down?

---

> ### Author Response · Authors · 2023-11-23
>
> Dear reviewer,
>
> We would like to thank the reviewer for their extensive feedback and careful reading of our manuscript.  Below, we respond in order of your questions, but before doing so, we would like to point to the **additional real-data experiment** motivated by your feedback about the experimental validation.  Additionally, we have also removed the vBF material as per your recommendation in order to make better use of the available space to unpack certain concepts e.g. the predictive plug-in through an illustrating example.
>
> > Why does L have different parameters in (10) vs (11)?
>
> This is an inconsistency on our side, thank you. We have fixed this in the revised manuscript. We tried to denote ELBOs where a parametric form for the approximation was not assumed yet using $\mathcal{L}(q)$.
>
> > Just below (14), you point out that as a result of the approximations it is no longer a valid ELBO. This seems bad (although not terminal), and I was surprised this issue is not referenced or addressed elsewhere. If this is being optimized end-to-end, and the objective is not a true bound, then you cannot make guarantees on what you are optimizing. Can the authors comment on this.
>
> Thank you for asking -- by not valid, we meant that it would not be an exact ELBO of the marginal likelihood.  While this approximation does break the bound, we are doing so for the sake of tractability; one nice intuitive feature of the plug-in approximations used is that in the linear and Gaussian case they are exact.  We also want to point out many existing works that are based around approximating difficult to manage terms in an ELBO/log-marginal likelihood with ones that are easier, e.g. site approximations of expectation propagation, tractable approximations ELBO to circumvent stochastic gradients [Keeley et al 2020, Efficient non-conjugate Gaussian process factor models…], or even using Delta approximations [Wang and Blei. 2012. Nonconjugate variational inference].
>
> Additionally, we have evaluated the tightness empirically in Fig.1. using the Kalman smoother and particle filter as baselines for the example linear and nonlinear dynamical systems respectively, we can see even in the nonlinear dynamics case the inferred posterior is close to the particle filter posterior.  We have revised the figure legend and title to make this more apparent.
>
> > At the bottom of page 4, you say the following:
> The backward factorization will make gaining insight into the structure of the variational posterior analytically easier; whereas $L_F(q)$ is easier to use in an amortized inference setting where trajectories from the approximate posterior are drawn and used to form a stochastic approximation of the bound (Krishnan et al., 2016; Karl et al., 2016). Again, this is not obvious to me. Nowhere else in the paper is the interpretability of the ELBO considered. You also assert that
> LF
>  is easier to evaluate compared to
> LB
> , but then later on, you evaluate
> LB
> , so I don't understand what the difference is.
>
>
> Thank you for your question.  We could have made the point clearer; we were trying to convey that the forward factorization and its ELBO makes it natural to design an inference network that amortizes $q(z_t | z_{t-1})$; however, amortizing $q(z_t | z_{t+1})$ and then evaluating L_B would be comparatively more difficult since we would have to use a Monte Carlo approximation of  $E[p(z_t|z_{t-1})]$, and additionally the KL between those distributions would also require a further Monte Carlo approximation. We have tried to explain this further in the revised manuscript.  However, what we do evaluate is $\hat{\mathcal{L}}_B$, which after substitution of the plug-in predictive, does not require making a MC approximation of the KL term.  We have tried to expand this point more in the main text.
>
> > This method be applied to any model with exponential family potentials? Can one mix and match potential types? I’m thinking something like a Poisson-HMM would be a nice demonstrative example of this, and one where there are many readily available baselines you could compare to.
>
> That is correct. Thank you for your suggestion -- in the revised manuscript, we the real data example shows this by using a Poisson likelihood to model neural spike-trains.

---

> > ### Author Response · Authors · 2023-11-23
> >
> > > Q.5.: (Following on from Q.4.) Are the D-VAE, L-VAE and N-VAE all different model classes, or proposal classes? It would be interesting to compare the inference performance of some different models on some different applications. For instance, with sufficiently small discretization steps, an extended Kalman filter would solve the Van Der Pol oscillator problem. How does the inference performance of both methods break down as that timestep grows larger? Have you tested how close to an exponential family the empirical distributions are, and whether your inference objective learns models that have “better” approximate posteriors?
> >
> > We summarized existing VAE methods for SSM inference that amortize the forward conditional posterior into D-VAE, L-VAE, and N-VAE. They are all different specifications for the inference networks in terms of increasing complexity (diffusion, linear, nonlinear);  the generative model is kept fixed, but these are specifications on the capacity of the inference networks.
> >
> > Our proposed method is different to those *-VAEs in terms of the backward factorization while the latter use forward ones $q(z_{1:T} = p(z_1)  \Pi_{2}^T q(z_t | z_{t-1})$ and are trained by evaluating the forward ELBO. The advantage of the backward factorization over the forward ones is the additional analytical tractability gained.
> >
> > > Can the authors clarify why natural gradient steps are preferable? What would happen if you didn’t use NG, and which other methods can use NG?
> >
> > Natural gradients here are preferable here because we are working solely with exponential family distributions, so natural gradients are no more costly to evaluate in this case than gradients in some other coordinate system, yet it offers a much more favorable ascent direction of the parameters in the space of probability distributions.
> > [Salimbeni, et al· 2018, Tan et al. 2021, Analytic natural gradient...]
> >
> >
> > >Q.7.: How computationally expensive is each method? You say that the Lagrange step is parallelizable, but the autoencoding variant almost certainly isn’t for sequence data (unless you use something like S4 as the RNN).
> >
> > The Lagrange variant is parallelizable like you point out. By parallelizable, we meant that it avoids recursion which has to be done sequentially. Indeed the autoencoding variant (aBFS) is not, but it has the complexity linear in the length of the sequence much like many deep state-space inference modeling works, which is not less efficient than any RNN structured methods. Importantly, by amortization, it does not require optimization in the inference mode. A unique feature of the autoencoding variant is that it can be easily tailored as a filtering method by removing the backward encoder. We have added a real data example in the revision to demonstrate the filtering.
> >
> > >Q.8.: I think I understand the constrained optimization step, but if the authors could clarify: Instead of optimizing all the parameters freely, you switch to an EM-like alternating optimization, where the M-step looks like a constrained optimization. This allows you to decouple the timesteps by essentially treating some of the summary statistics as fixed. Does this coordinate-wise method have any drawbacks?
> >
> > That is correct for the Lagrange multiplier method; we hold the Lagrange multipliers fixed and optimize the parameters of the variational approximation fully, then take a gradient step on the Lagrange multipliers in a dual ascent-like fashion.  Similar to dual ascent methods in optimization, we found that convergence in the dual variable space could be slow -- we mention this as a weakness in the experiments, and also consider future work to accelerate convergence in the conclusion.
> >
> > >Q.9.: This is a relatively core question, but can the authors concretely re-state why we should consider a backward factorization of the proposal? Specifically which step cannot be achieved if we use a forward factorization? And where does each method break down?
> >
> > The strength of using the backward factorization is in the analytical simplifications made possible by parameterizing $q(z_{t-1} | z_t) \propto p(z_t | z_{t-1}) q(z_{t-1})$ -- we do not really want sample backwards in time, the goal was to develop a way of factorizing the ELBO that avoided sampling entire trajectories from the approximate posterior, and the backward factorization was an instrument to do so.
> >
> > Thank you again for your valuable feedback, it was immensely helpful in restructuring the manuscript and making the story more cohesive.  We feel that by replacing the vBF material, which was less connected to the main story, with more intuitive explanations and a more complex experimental evaluation as per your suggestions has resulted in a more well positioned paper.   We hope that you could consider adjusting your score if you find it warranted given the extra improvements.  We greatly appreciate the depth and value of your comments and suggestions.
> >
> > Thank you again for your time,
> > The authors

---

### Official Review · Reviewer_vygh · 2023-10-31

**Soundness:** 3 good
**Presentation:** 3 good
**Contribution:** 2 fair
**Rating:** 5
**Confidence:** 3

**Summary:**

This paper develops a number of algorithms for smoothing of exponential family dynamical systems. The main contributions are summarized as follows.

First, the paper introduces a prior-parameterized backward factorization to the smoothing posterior. This factorization $ q(z_t|z_{t-1}) $ includes a model of the dynamics $ p_\theta(z_{t+1}|z_t)$  so that it factors similarly to the true posterior, and additionally allows for linear scaling with the number of timesteps without the assumption of independence (as in mean field approximation). When using this backward factorization, one obtains a convenient and simplified expression for the ELBO, which includes terms that are KLs of expectations through conditionals, as opposed to KLs of expectations. One thing that was unclear to me was the claim that "The backward factorization will make gaining insight into the structure of the variational posterior analytically easier". Could this be elaborated on? Additionally, to facilitate tractability, all terms in this factorization were replaced by their closest exponential family approximations.

Second, the authors develop a smoothing algorithm (dBFS) that allows for parameters to be learned in parallel with the addition of lagrange multipliers which constrains $\mu_t^- = E_{q_{t-1}}[\mu_\theta(z_{t-1})]$, i.e. consistency of the parameters between timesteps.

Third, by noting the relationships between the natural parameters and the Lagrange multipliers (Eq. 20), the authors introduce an amortized smoothing algorithm (aBFS) that introduces function approximators to learn the functions present in these relationships, leading to a VAE-style algorithm for smoothing. The authors also show how this algorithm can be applied for sequential inference for more practical applications in Sec 3.3 via vBF.

**Strengths:**

To begin, I thought this approach was extremely elegant. Utilizing the insights from Section 2.4, the authors compared the forward and backward factorization approaches and gave interesting insights as to when one might be preferred over the other. The authors also contextualized their work well in the short but informative Section 4. Overall, the paper is explained well, and the ideas were clear to understand. It was hard to evaluate how significant this work is due to the limited experiments (discussed below).

**Weaknesses:**

The Experimental section I think is both slightly difficult to understand and also not as compelling as the rest of the paper. There was a single experiment for dBFS, as well as a pendulum experiment for aBFS, as well as an experiment on smoothing chaotic dynamics that used aBFS and vBF. From this set of experiments, it was hard to get a good understanding of how 1) dBFS compares with aBFS, 2) how useful aBFS really is relative to the baselines, and 3) how aBFS and vBF would scale to more complex problems. I think there is a lot of room for improvement here, and thus think the paper has a lot of unrealized potential in its current form.

**Questions:**

-Is it possible to elaborate on what it means to be a "plug-in" predictive distribution? Are we just saying that we will take the thing that is supposed to be $q^-$ and plug in $E_{q(z_{t-1})}[p_\theta(z_t|z_{t-1})]$ instead? Is this different than a posterior predictive distribution?

-Is it possible to state the number of total parameters that each of the models in the experiments had? I think having significantly fewer parameters relative to something like L-VAE could be interesting to demonstrate the models effectiveness.

---

> ### Author Response · Authors · 2023-11-23
>
> Dear reviewer,
>
> Thank you for taking the time to go through our manuscript, we appreciate your valuable feedback. We will try to address your questions in order.  In particular, we would like to point to the **additional real-data experiment** motivated in part by the feedback you provided about the experimental validation; additionally, we have also added **further intuition/motivation behind the plug-in predictive** and a walk through of how it is exact in the linear gaussian case. Motivated by other reviewer suggestions, we have also removed the vBF section in favor of using the additional space to add more details and have a more cohesive story. Below, we have tried to answer your questions.
>
> > Is it possible to elaborate on what it means to be a “plug-in” predictive distribution? Is this different than a posterior predictive distribution?
>
> Yes, exactly. It was intended as an acting posterior predictive distribution for a single-step into the future. We name it as we do “plug-in” $\bar{q}$ as an approximation to $E_{q_t}[p(z_t | z_{t-1})]$ (the best exponential family approximation as measured by the forward KL).
>
> > is it possible to state the number of total parameters that each of the models in the experiments had?
>
> Certainly, we have added these in the experiments section.
>
>
> >  One thing that was unclear to me was the claim that "The backward factorization will make gaining insight into the structure of the variational posterior analytically easier". Could this be elaborated on?
>
> Thank you for the question, our wording here could be improved a lot, and we hope revisions to the manuscript clear it up.  Since the backward factorization leads to the KLs of expected distributions, our insight was that the KL between exponential family distributions can usually be calculated in closed form, so that combined with the subsequent approximation we would be better equipped to gain analytical insights from the approximate ELBO.
>
> The explicit backward conditional distribution of backward factorization leads to analytical backward information propagation as an additive contribution to the posterior natural parameter, unlike other methods that use deep NN amortization e.g. bidirectional RNN or transformer that blackboxed and mixed the backward information propagation with forward message and likelihood contribution.
>
>
> > 3) how aBFS and vBF would scale to more complex problems. I think there is a lot of room for improvement here, and thus think the paper has a lot of unrealized potential in its current form.
>
> Thank you for this question, it has helped motivate us to apply aBFS to real data in order to validate its efficacy.  We hope the additional experiment applying aBFS to analyzing a more complex real dataset will help position the paper better.
>
> Again we would like to thank you for your valuable feedback, which we feel has helped to greatly improve the manuscript through additional discussion and applying aBFS to a real dataset to demonstrate its efficacy.  We hope that if we have addressed your concerns, you might increase your score to reflect that.
>
> Thank you again for your time,
>
> The authors

---

### Meta-Review · Area_Chair_sxTT · 2023-12-04

**Metareview:**

This is an interesting paper that provides a novel structured variational approximation to exponential family dynamical systems. This enables us to consider non-Gaussian state space models, given accompanying algorithms such as their backward factorized smoother. I quite like this idea and find it elegant and interesting; several reviewers had positive sentiments but ultimately found that the results were not quite convincing enough to generate more excitement. I think the paper would benefit from more clear evidence or well designed examples to show what we gain by using this fairly sophisticated machinery, and also when this fails or is limited. I encourage the authors to take into account these detailed reviewer comments in a revision

**Justification For Why Not Higher Score:**

borderling reviews erring on negative

**Justification For Why Not Lower Score:**

na

---

### Decision · Program_Chairs · 2024-01-16

Reject